# Horizon Reduction Makes RL Scalable

**Seohong Park**[1]    **Kevin Frans**[1]    **Deepinder Mann**[1]
**Benjamin Eysenbach**[2]    **Aviral Kumar**[3]    **Sergey Levine**[1]
[1]University of California, Berkeley    [2]Princeton University    [3]Carnegie Mellon University
seohong@berkeley.edu

## Abstract

In this work, we study the *scalability* of offline reinforcement learning (RL) algorithms. In principle, a truly scalable offline RL algorithm should be able to solve any given problem, regardless of its complexity, given sufficient data, compute, and model capacity. We investigate if and how current offline RL algorithms match up to this promise on diverse, challenging, previously unsolved tasks, using datasets up to $1000\times$ larger than typical offline RL datasets. We observe that despite scaling up data, many existing offline RL algorithms exhibit poor scaling behavior, saturating well below the maximum performance. We hypothesize that the *horizon* is the main cause behind the poor scaling of offline RL. We empirically verify this hypothesis through several analysis experiments, showing that long horizons indeed present a fundamental barrier to scaling up offline RL. We then show that various horizon reduction[1] techniques substantially enhance scalability on challenging tasks. Based on our insights, we also introduce a minimal yet scalable method named SHARSA that effectively reduces the horizon. SHARSA achieves the best asymptotic performance and scaling behavior among our evaluation methods, showing that explicitly reducing the horizon unlocks the scalability of offline RL.

Code: https://github.com/seohongpark/horizon-reduction

## 1 Introduction

Scalability, the ability to consistently improve performance with more data and compute, is at the core of the success of modern machine learning algorithms, across natural language processing (NLP), computer vision (CV), and robotics. In this work, we are interested in the scalability of *offline reinforcement learning (RL)*, a framework that can leverage large-scale offline datasets to learn performant policies. While prior works have shown that current offline RL methods scale to *more* (but not necessarily harder) tasks with larger models and datasets [49, 86], it remains unclear how RL scales with data to *more challenging* tasks, especially those that require more complex, longer-horizon sequential decision making.

Our main question, posed informally is:

*To what extent can current offline RL algorithms solve complex tasks*
*simply by scaling up data and compute?*

In principle, a truly scalable offline RL algorithm should be able to master *any* given task, *no matter how complex and long-horizon it is*, given a sufficient amount of data (of sufficient coverage), compute, and model capacity. Studying how current offline RL algorithms live up to this promise is important, because it will tell us whether we are ready to scale existing offline RL methods, or if we must further improve offline RL algorithms before scaling them.

---

[1]Throughout this work, we use the term "horizon reduction" to refer to techniques that reduce the *effective* decision horizon, such as $n$-step returns and hierarchical policies.

39th Conference on Neural Information Processing Systems (NeurIPS 2025).

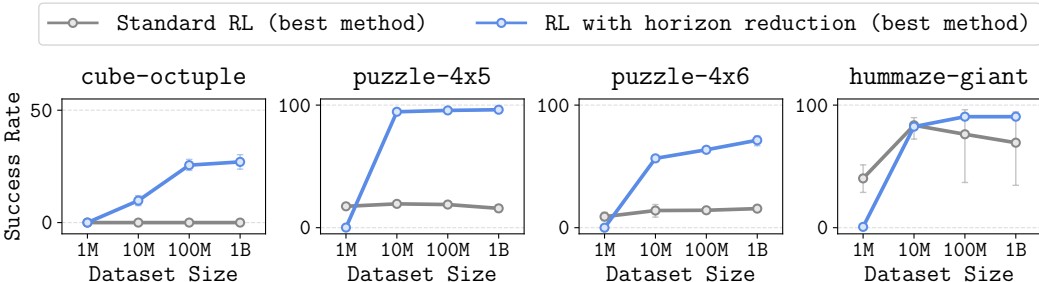

Figure 1: **Horizon reduction makes RL scalable.** Standard offline RL methods struggle to scale on highly challenging tasks, not improving performance with more data. We show that this is mainly because the *long horizon* can fundamentally inhibit scaling, and that horizon reduction techniques unlock the scaling of offline RL.

To answer this question, we generate large-scale datasets for tasks that require highly complex, long-horizon reasoning, and study how current offline RL algorithms scale with data. Specifically, on complex simulated robotics tasks across diverse domains in OGBench [75], we collect a dataset with up to **one billion** transitions for each environment, which is $1000\times$ larger than standard 1M-sized offline RL datasets used in prior work [21, 75]. To isolate the fundamental sequential decision-making capabilities of RL algorithms, we also idealize environments by removing other potential confounding factors, such as visual representation learning. In these controlled yet challenging environments, we evaluate the performance of state-of-the-art offline RL algorithms while varying the amount of data.

We observe that many existing offline RL algorithms struggle to scale, even with orders of magnitude more data in these idealized environments. Specifically, we show algorithms such as IQL [47], SAC+BC (Appendix E.1), CRL [18], and FQL [76] often either completely fail to solve complex tasks, or require an excessive amount of compute and model capacity to reach even moderate performance. Their performance often saturates far below the maximum possible performance (Figure 1), especially on complex, long-horizon tasks, suggesting that there exist scalability challenges in offline RL.

We hypothesize that the reason behind this poor scaling is due to **the curse of horizon** in both value learning and policy learning. In value learning, we argue that the temporal difference (TD) learning objective used in many offline RL algorithms has a fundamental limitation that inhibits scaling to longer horizons: biases (errors) in the target Q values *accumulate over the horizon*. Through controlled analysis experiments, we show that this bias accumulation is strongly correlated with poor performance. Moreover, we show that increasing the model size or adjusting other hyperparameters, does *not* effectively mitigate this issue, suggesting that the horizon fundamentally hinders scaling. In policy learning, we argue that the complexity of the mapping between states and optimal actions in long-horizon tasks poses a major challenge, and support this claim with experiments.

We then demonstrate that methods that explicitly reduce the value or policy horizonsexhibit substantially better scaling (Figure 1). For example, we show that even simple techniques to reduce the value horizon, such as $n$-step returns, can substantially improve scaling curves and even asymptotic performance. Based on our insights, we also propose a minimal yet scalable RL method called **SHARSA** that reduces both the value and policy horizons. Our method relies only on simple objectives that do *not* require excessive hyperparameter tuning, such as SARSA and behavioral cloning, while effectively reducing the horizon. Despite the simplicity, we show that SHARSA generally exhibits the best scaling behavior and asymptotic performance among our evaluation methods.

**Contributions.** Our main contributions are threefold. First, through our 1B-scale data-scaling analysis, we empirically demonstrate that many standard offline RL algorithms scale poorly on complex, long-horizon tasks. Second, we identify the *horizon* as a main obstacle to RL scaling, and empirically show that horizon reduction techniques can effectively address this challenge. Third, we propose a simple method, SHARSA, that exhibits strong asymptotic performance and scaling behavior.

## 2 Related work

**Offline RL.** Offline RL aims to train a reward-maximizing policy from a static dataset without online interactions [55]. The main challenge in offline RL is to maximize rewards while staying close to the dataset distribution to avoid distributional shift. Previous works have proposed a number of techniques to address this challenge based on behavioral regularization [22, 76, 91, 99], conservatism [48],

weighted regression [77, 78, 97], in-sample maximization [25, 47, 100], uncertainty minimization [3, 71], one-step RL [7, 18], model-based RL [43, 102, 103], and more [10, 31, 39, 40, 53, 84, 96]. Among these methods, we mainly consider three distinct representative model-free algorithms that have been reported to achieve state-of-the-art performance on standard benchmarks [75, 76, 92], IQL [47], SAC+BC (Appendix E.1), and CRL [18], as the main subject of our scaling analysis. We leave the scaling study of offline model-based RL for future work.

**Scaling RL.** Prior work has studied the scalability of RL algorithms in various aspects. Many previous works focus on scaling *online* RL to solve more diverse tasks with larger models [29, 30, 70], more compute [81], and parallel simulation [16, 24, 58, 85]. More recently, several works have also explored the scaling of online, on-policy RL on language tasks [38, 94]. Unlike these works that study online RL, we focus on the scalability of *offline* RL algorithms.

Many prior works on scaling offline RL focus on scalability to *more* tasks by training a large, multi-task agent that is capable of solving more diverse (but not necessarily harder) tasks [8, 11, 49, 54, 80, 86]. Unlike these works, we focus the ability to solve more *challenging* tasks that require highly complex sequential decision making, given more data and compute. This is analogous to scalability along the "depth" axis, as opposed to the "width" axis that the prior works have explored. This is an important, complementary axis to study, as it will let us know whether offline RL is currently bottlenecked by the amount of data and compute, or the fundamental learning capabilities of algorithms. A closely related work is Park et al. [73], which shows that poor policy extraction and generalization can bottleneck the scaling of offline RL. We study the scalability of offline RL to more complex (in particular, longer-horizon) tasks when these bottlenecks are removed, with the use of more expressive policy classes and with datasets $100\times$ as large as this prior work. This makes our study distinct from and complementary to the challenges that Park et al. [73] highlight.

**Horizon reduction and hierarchical RL.** In this work, we identify the horizon length as one of the main factors that inhibit the scaling of RL. Prior works have developed diverse techniques to reduce the effective horizon with multi-step or hierarchical value functions [1, 5, 14, 56, 66, 87, 90, 106], hierarchical policy extraction [26, 62, 72], or high-level planning [17, 19, 35, 36, 44, 45, 57, 69, 74, 82]. While these works in hierarchical RL have mainly focused on exploration [68], representation learning [67, 72], and planning [17], we focus on *scalability*, showing that horizon reduction mitigates bias accumulation and unlocks the scaling of offline RL. In this work, we also propose a new, minimal method (SHARSA) to reduce the horizon. SHARSA is related to previous hierarchical methods that use rejection sampling for subgoal selection [1, 33, 64]. Inspired by these works, SHARSA uses a minimal set of techniques (*e.g.*, flow behavioral cloning and SARSA) that address the horizon issue in a scalable manner (see Section 6.1 for further discussions).

## 3 Experimental setup

**Problem setting.** We aim to understand the degree to which current offline RL methods can solve complex tasks simply by scaling data and compute. In particular, we are interested in the capabilities of offline RL algorithms to solve challenging tasks that require *complex, long-hoziron* sequential decision-making given enough data. To this end, we focus on the *offline goal-conditioned RL* setting [75], where we want to train agents that are able to reach any goal state from any other initial state in the fewest number of steps, from a static, pre-collected dataset of behaviors. This problem poses a substantial learning challenge, as the agent must learn complex, long-horizon, multi-task behaviors purely from binary sparse rewards and an unlabeled (reward-free) dataset. We note that although we mainly focus on goal-conditioned tasks in this work, our claims are not limited to goal-conditioned settings (Appendix D).

Formally, we consider a controlled Markov process defined as $\mathcal{M} = (\mathcal{S}, \mathcal{A}, \mu, p)$, where $\mathcal{S}$ is the state space, $\mathcal{A}$ is the action space, $\mu(s) \in \Delta(\mathcal{S})$ is the initial state distribution and $p(s' \mid s, a) : \mathcal{S} \times \mathcal{A} \to \Delta(\mathcal{S})$ is the transition dynamics kernel. Here, $\Delta(\mathcal{X})$ denotes the set of probability distributions on space $\mathcal{X}$, and we denote placeholder variables in gray. We denote the discount factor as $\gamma$. The dataset $\mathcal{D} = \{\tau^{(n)}\}_{n \in \{1, 2, \ldots, N\}}$ consists of $N$ length-$H$ state-action trajectories, $\tau = (s_0, a_0, s_1, a_1, \ldots, s_H)$. We assume that these trajectories are collected in an unsupervised, task-agnostic manner.

**Environments and datasets.** We employ four highly challenging offline goal-conditioned RL tasks in robotics from the OGBench task suite [75]. Among these tasks, `cube` involves sequential pick-and-place manipulation of multiple cube objects, `puzzle` involves solving a combinatorial puzzle called "Lights Out"[2] with a robot arm, and `humanoidmaze` involves whole-body control of a humanoid agent to navigate a given maze. These OGBench tasks provide multiple variants with varying levels of difficulty, and we employ the hardest tasks (`cube-octuple`, `puzzle-{4x5, 4x6}`, and `humanoidmaze-giant`) to maximally challenge offline RL algorithms. To our knowledge, no current offline RL algorithm has been reported to achieve non-trivial performance (*i.e.*, non-zero performance on most evaluation goals) on these hardest tasks with the original OGBench datasets [75].

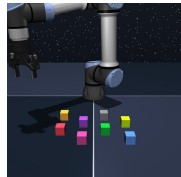 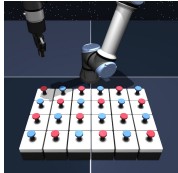

cube-octuple     puzzle-4x6

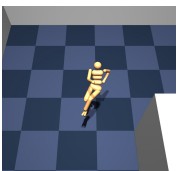 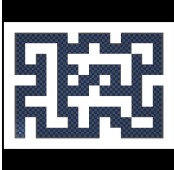

humanoidmaze-giant

On these environments, we generate up to **1B** transitions using the scripted policies provided by OGBench. These datasets consist of "play"-style [62] task-agnostic trajectories to ensure sufficient coverage and diversity (see also the discussion about dataset coverage in Section 4). Specifically, they contain trajectories that randomly navigate the maze (`humanoidmaze`), sequentially perform random pick-and-place (`cube`), or press random buttons (`puzzle`). These task-agnostic, unsupervised datasets conceptually resemble unlabeled Internet-scale data used to train vision and language foundation models. We also note that our 1B-sized datasets contain about 1M trajectories and 10M atomic behaviors in manipulation environments, which is similar or even larger than one of the largest robotics datasets to date [13].

**Idealization.** To isolate the core sequential decision-making capabilities of RL from other confounding factors, such as challenges with visual representation learning, distributional shift, and data coverage, we idealize environments and tasks in our analysis experiments. While these challenges are certainly important in practice, our rationale is to first understand how current offline RL algorithms can solve highly challenging tasks in an idealized, controlled setting with near-infinite data.

Specifically, we employ low-dimensional state-based observations with oracle goal representations to alleviate challenges in visual representation learning. We also remove some evaluation goals that may require out-of-distribution generalization to ensure all tasks remain in-distribution. Finally, we ensure that the datasets have sufficient coverage and optimality, by verifying that our data-collecting script enables achieving near-perfect performance on the same environment with fewer objects (see Section 4 for the full discussion). We refer to Appendix F for the details.

**Methods we evaluate.** In this work, we mainly consider three performant, widely-used offline model-free RL algorithms across different categories: IQL, CRL, and SAC+BC. IQL [47] is based on in-sample maximization, CRL [18] is based on contrastive learning and one-step RL [7], and SAC+BC (Appendix E.1) is based on behavioral regularization [22, 99]. Additionally, we employ flow behavioral cloning (flow BC) [6, 12] to understand the scalability of behavioral cloning as well. Due to high computational costs, we use four random seeds in our scaling experiments (unless otherwise noted), and report 95% confidence intervals with shaded areas in the plots. A full description and implementation details of the algorithms are provided in Appendices E and F.

## 4 Standard offline RL methods struggle to scale

We now evaluate the degree to which four standard offline RL methods (flow BC, IQL, CRL, and SAC+BC) can solve the four challenging tasks by simply scaling up data. Figure 3 shows the scaling plots of the four methods with 1M, 10M, 100M, and 1B-sized datasets (see Figure 13 for the full training curves). These methods are trained for 5M steps with [1024, 1024, 1024, 1024]-sized multi-layer perceptrons (MLPs).

In aggregate, our results show that **none** of these four standard offline RL methods are able to solve all four tasks, even with the largest 1B-sized datasets. Notably, all of them completely fail on the hardest `cube-octuple` task. Moreover, their performance often quickly plateaus well below the

---

[2]https://en.wikipedia.org/wiki/Lights_Out_(game).

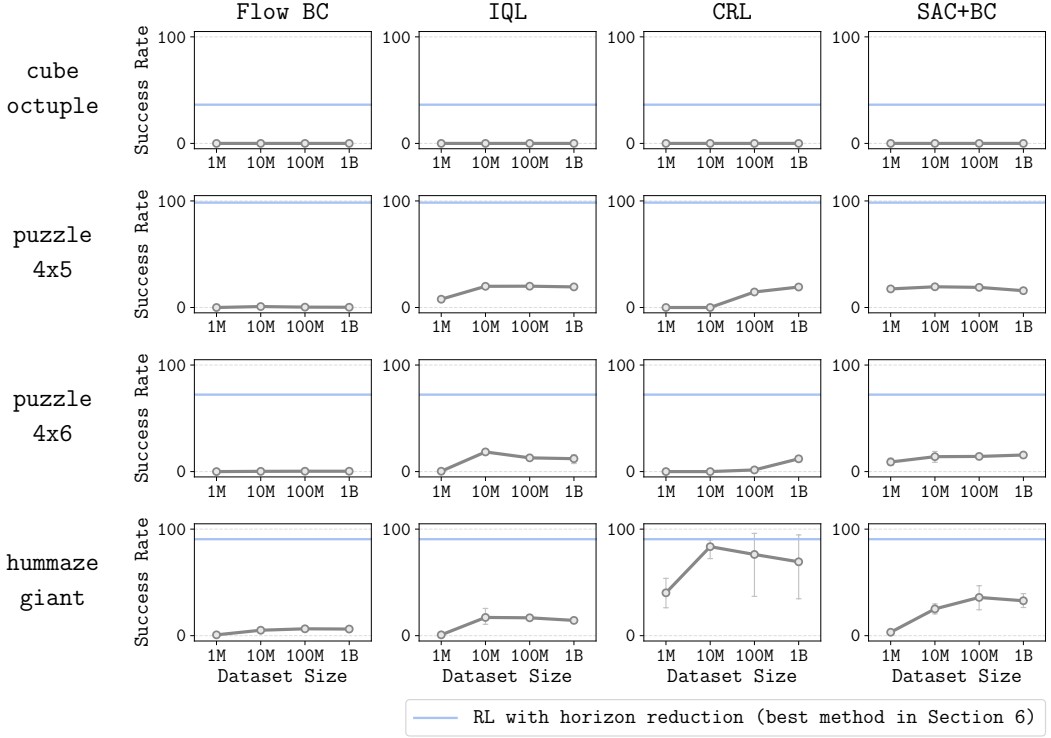

Figure 3: **Standard offline RL methods struggle to scale on challenging tasks.** We train four offline RL methods with 1M, 10M, 100M, and 1B data on four complex, long-horizon tasks. However, even with 1B data, their performance often saturates far below the maximum performance (100%).

optimal success rate (*i.e.*, 100%), despite scaling up data. In other words, these standard offline RL methods struggle to scale on these tasks.

A keen reader may already have several questions about this result. Before proceeding further, we first address those potential questions.

**Q: How do you know these tasks are solvable with the given datasets?**

**A:** We will see in Section 6 that it is indeed possible to solve these tasks, or at least achieve non-trivial performance (denoted in blue in Figure 3). In Appendix A, we also show that these algorithms can solve the same tasks with fewer objects, using datasets collected by the same scripted policy. This verifies that the dataset *distribution* (induced by the scripted policy) provides sufficient coverage to learn a near-optimal policy.

**Q: Have you tried further increasing the model size?**

**A:** A natural confounder in the results above is the model size. To understand how this affects performance, we train SAC+BC, the best method on `cube-double` and `puzzle-4x4` (Figure 10), using up to $35\times$ larger models with 591M parameters, on the largest 1B datasets. Figure 4 shows the training curves. The results suggest that while using larger networks can improve performance on some tasks to some degree, this alone is not sufficient to master the tasks, especially the hardest `cube-octuple` task. Moreover, the performance often saturates (or sometimes degrades) despite using larger models. In Appendix B, we provide more ablations with different architectures (residual MLPs and Transformers), which show similar trends.

While an even larger network with a smaller learning rate[3] might further improve performance (which unfortunately we could not afford, as 591M models already require 8 days of training), we are interested in performance within a reasonably bounded total compute budget. If an algorithm is unable to achieve good performance within a practical amount of compute, we deem it a challenge in scalability. In contrast, in Section 6, we will show that horizon reduction techniques enable achieving

---

[3]We already use a decreased learning rate for the largest 591M model; see Appendix B for the details.

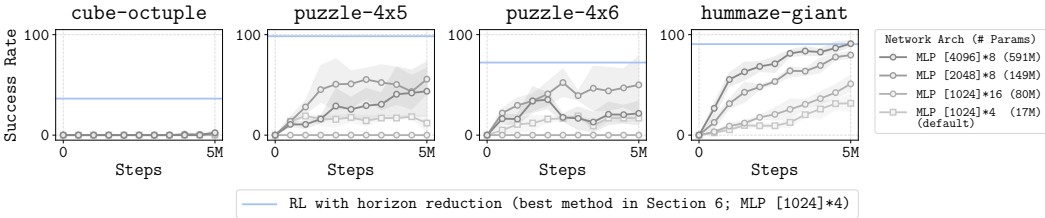

Figure 4: **Increasing model capacity alone is not sufficient to master the tasks.**

significantly better scaling behavior and asymptotic performance (denoted in blue in the above figure) even with the original $[1024] \times 4$-sized models.

**Q: Are you sure this isn't just a hyperparameter or design choice issue?**

**A:** While there is always a possibility of achieving better performance with better hyperparameters, despite our extensive efforts in adjusting hyperparameters and design choices, we were unable to achieve promising scaling results with these methods. In Appendix B, we present **9** ablation studies on policy classes (Gaussian and flow policies), network architectures (MLPs and Transformers), value ensembles, regularization techniques, learning rates, target network update rates, batch sizes, and gradient steps, showing that **none** of these changes substantially improves scalability or asymptotic performance across the board.

## 5 The curse of horizon

Why do current offline RL methods exhibit poor scaling behavior on these challenging tasks? In the previous section, we observed that adjusting model sizes or other hyperparameters does *not* effectively improve scaling on complex, long-horizon tasks, even though they scale on simpler tasks (see Figure 10). This suggests that there may exist a fundamental obstacle that inhibits the scaling of offline RL. We hypothesize that this obstacle is the **horizon**. In this section, we discuss and analyze *the curse of horizon* along two orthogonal axes: value and policy.

### 5.1 The curse of horizon in value learning

Many offline RL algorithms train Q functions via temporal difference (TD) learning. Unfortunately, the TD learning objective has a fundamental limitation: at any gradient step, the prediction target that the algorithm chases is *biased* [89], and these biases *accumulate* over the horizon. Such biases do not exist (or at least they do not accumulate) in many scalable supervised and unsupervised learning objectives, such as next-token prediction. As such, we hypothesize that the presence of bias accumulation in TD learning is one of the fundamental causes behind the poor scaling result in Section 4. This hypothesis partly explains why CRL, which is not based on TD learning, achieves a significantly better asymptotic performance on `humanoidmaze-giant` in Figure 3.

**Didactic task setup.** We empirically validate this hypothesis by performing an analysis on a didactic task named `combination-lock` (Figure 5). This environment has $H$ states and two discrete actions. The states are linearly ordered, and each state has an "answer" action. The state order and answer actions are randomly chosen (and kept fixed) when instantiating the environment. The agent starts from the first state,

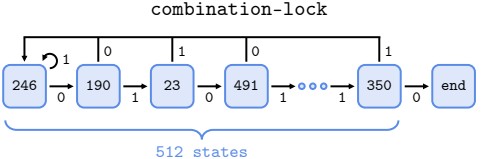

Figure 5: `combination-lock` with **H = 512.**

and whenever it selects the correct action, it moves forward by one step; otherwise, it is sent back to the first state. The agent always gets a reward of $-1$ at each step, except at the final (goal) state, where it gets a reward of $0$ and the episode terminates. Hence, the agent must memorize all $H$ answer actions to reach the goal.

To understand the effect of bias accumulation in deep TD learning, we evaluate two offline Q learning algorithms with different TD horizons: standard (1-step) DQN [65] and $n$-step DQN (see Appendix F.1 for details).[4] Note that the optimal Q functions for both algorithms are the same

---

[4]Although these are online RL algorithms, we can also use them for offline RL without modification, as our datasets have uniform coverage and thus do not require conservatism.

(under the optimal, full-coverage datasets) and thus have the same learning complexity, but the latter involves $n$ times fewer TD recursions. To compare the maximum possible performance of these two algorithms in a fair way, we employ two types of datasets that have uniform coverage of length-$\{1, n\}$ trajectory segments, evaluate each method on both datasets, and select the best one for each method. In this experiment, we do not use a discount factor (*i.e.*, $\gamma = 1$) as the task has a finite horizon. We refer the reader to Appendix F.1 for the full experimental details.

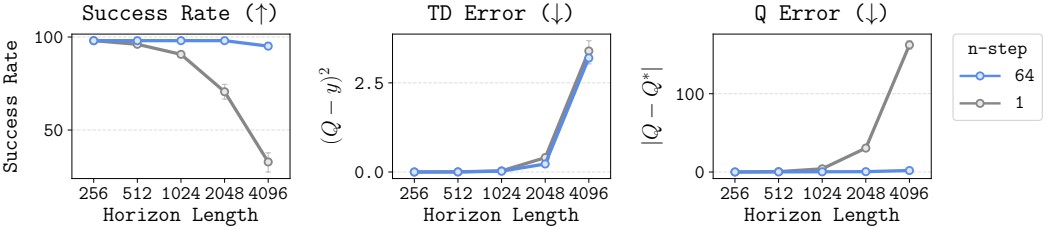

Figure 6: **1-step TD learning suffers bias accumulation (*i.e.*, high Q errors).**

We train $1$-step and $64$-step DQN on `combination-lock` with different horizon lengths, ranging from $H = 256$ to $H = 4096$. First, we measure their performance. The first plot in Figure 6 shows that the performance of 1-step DQN drops faster than that of 64-step DQN as the horizon increases.

Next, we measure two metrics: the TD error and the Q error. The *TD error* measures the difference against the TD target $y$, and the *Q error* measures against the ground-truth Q value $Q^*$. The results are presented in the second and third plots in Figure 6. They show that 1-step DQN has significantly larger Q errors than 64-step DQN, even though they have similar TD errors. Since the Q error corresponds to compounded error in the learned Q function, this strongly suggests that bias accumulation happens in practice, and that it can substantially affect performance on long-horizon

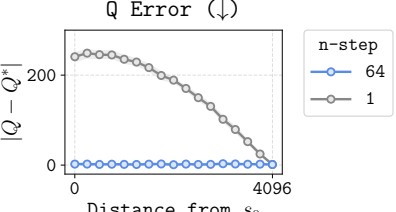

Figure 7: **Biases accumulate.**

tasks. We further corroborate this point by measuring how Q errors vary across state positions in a single episode. Figure 7 shows that Q errors indeed become larger as the distance from the end increases.

Then, is it possible to fix error accumulation in 1-step TD learning by tuning hyperparameters, or is it a fundamental limitation of deep TD learning? As in Section 4, we adjust diverse hyperparameters, such as the model size, learning rate (LR), and target network update rate (TUR), and present the ablation results in Figure 8. The results suggest that simply increasing the model size or decreasing LR or TUR provides limited or no improvement in both performance and bias accumulation (measured by Q errors). This matches the observation in Section 4. In contrast, 64-step DQN achieves significantly better performance and Q errors, even with the default-sized network. This suggests that error accumulation over the horizon may be a fundamental factor that obstructs scaling up TD learning.

Of course, there is a possibility that under certain hyperparameter settings, such as with a very low learning rate and a much larger network, 1-step DQN might be able to converge to the optimal policy on long-horizon tasks. While it is impossible to experimentally eliminate this possibility entirely, our results do suggest 1-step TD learning *scales poorly* in horizon, in the sense that it may require an excessive amount of compute, model capacity, and the practitioner's time to achieve good performance. In contrast, our results show that techniques that explicitly reduce the effective horizon, such as $n$-step returns (as one example), can potentially be much more effective in addressing this issue.

### 5.2 The curse of horizon in policy learning

Orthogonal to the bias accumulation issue in value learning discussed in the previous section, the policy may also independently suffer from the curse of horizon. This is because, even when the value function is perfect, the policy still needs to *fit* the mapping between states and optimal actions prescribed by the Q function, where this mapping can be increasingly complex as the horizon becomes longer. For example, in the goal-conditioned setting, the mapping between the optimal actions and distant goals can be highly complex [75], as it depends on the entire topology of the state space.

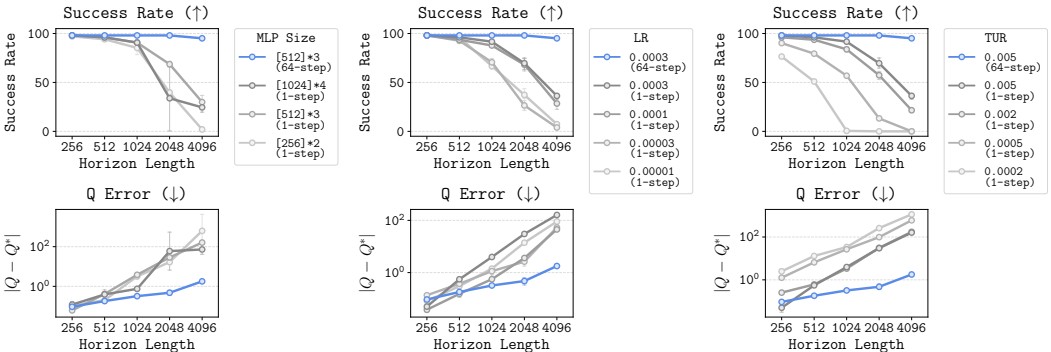

Figure 8: **Regardless of hyperparameters, 1-step TD learning struggles to handle a long horizon.**

Analogous to $n$-step returns in value learning, we can mitigate the curse of horizon in policy learning by reducing the effective horizon using a *hierarchical* policy [26, 62, 72, 75]. For example, we can decompose a goal-conditioned policy $\pi(a \mid s, g)$ into a high-level policy $\pi^h(w \mid s, g)$ that outputs a subgoal $w$, and a low-level policy $\pi^\ell(a \mid s, w)$ that outputs actions given the subgoal. Since the complexity of the individual hierarchical policies is often (much) lower than that of the flat (*i.e.*, non-hierarchical) policy [72], this can lead to a policy that both performs and *generalizes* better [75]. This is akin to how chain-of-thought reasoning [98] improves the performance of language models, which shows that decomposing a problem into multiple simpler subtasks is more effective than producing an answer directly. While we do not perform a separate didactic experiment for this point (as it is relatively well studied and analyzed in prior work [26, 72, 75]), we will empirically demonstrate how hierarchical policies can substantially improve the scalability of offline RL in challenging environments in the next section.

## 6   Horizon reduction makes RL *scale* better

Based on the insights in Section 5, we now apply value and policy horizon reduction techniques to our four challenging benchmark tasks, and evaluate how they improve the scalability of offline RL.

### 6.1   Horizon reduction techniques

As discussed in the previous section, there are two orthogonal axes of horizons in RL: the value horizon (Section 5.1) and policy horizon (Section 5.2). In our experiments, we consider four representative techniques that reduce one or both types of horizons. We refer to Appendix E.2 for the full details.

**Value horizon reduction.** For value horizon reduction, we consider $n$**-step SAC+BC**, a variant of SAC+BC with $n$-step TD updates, analogous to $n$-step DQN in Section 5.1. This method reduces the value horizon, but not the policy horizon, as it learns a flat policy.

**Policy horizon reduction.** We consider two techniques that reduce the policy horizon, but not the value horizon. **Hierarchical flow BC (hierarchical FBC)** trains a hierarchical policy ($\pi^h$ and $\pi^\ell$) with flow behavioral cloning, without performing RL. **HIQL** [72] trains a flat value function with goal-conditioned IQL, but extract a hierarchical policy from it. These methods will tell us the degree to which having a hierarchical policy *alone* can improve performance.

**Value *and* policy horizon reduction.** We can reduce both the value and policy horizons with full-fledged hierarchical RL. While there are several approaches that perform full hierarchical offline RL with a (potentially complex) high-level *planner* (Section 2), there exist only a handful of planning-free methods that reduce *both* the value and policy horizons [1, 33, 64]. Since these methods are either based on (less scalable) variational autoencoders or recurrent networks [1, 64], or only applicable to language-based tasks [33], we propose a new method called **SHARSA** in the following section.

### 6.2   SHARSA: a minimal, scalable offline RL method for horizon reduction

We propose a simple, scalable offline RL method that reduces both the value and policy horizons for continuous control. Our main goal here is, rather than designing a completely novel technique

that achieves state-of-the-art performance, to empirically demonstrate how reducing both types of horizons improves scalability, even with otherwise simple ingredients.

The main challenge with full hierarchical offline RL (*i.e.*, value and policy horizon reduction) is *high-level policy extraction*: learning a subgoal policy $\pi^h(w \mid s, g)$ that maximizes values while not deviating too much from the data distribution. For low-level or flat policies (whose output space is $\mathcal{A}$), policy extraction is typically best done by reparameterized gradients in the action space [73] (*e.g.*, DDPG+BC [22, 73]). However, the same technique does not necessarily work for high-level policies (whose output space is $\mathcal{S}$), since first-order gradient information in the *state* space may not be semantically meaningful (*e.g.*, the button states of puzzle are discrete, and thus first-order gradients in the state space are not even well-defined).

To address this challenge, we employ *rejection sampling* [9, 31, 33, 64] with an expressive *flow* policy [2, 6, 59, 61, 76] for high-level policy extraction: we first sample $N$ subgoals from a high-level flow BC policy $\pi_\beta^h$ and pick the best one based on a high-level (goal-conditioned) value function $Q^h$:

$$\pi^h(s, g) \stackrel{d}{=} \underset{w_1,\ldots,w_N : w_i \sim \pi_\beta^h(w|s,g)}{\arg\max} Q^h(s, w_i, g), \tag{1}$$

where $\stackrel{d}{=}$ denotes equality in distribution. This is beneficial because it does not use first-order information (unlike reparameterized gradients) while leveraging the expressivity of a flow policy [73, 76]. For the value function $Q^h$ in Equation (1), we employ high-level SARSA [89] for simplicity, which trains behavioral value functions with the following losses:

$$L^V(V^h) = \mathbb{E}\Big[D\big(V^h(s_h, g), \bar{Q}^h(s_h, s_{h+n}, g)\big)\Big], \tag{2}$$

$$L^Q(Q^h) = \mathbb{E}\Big[D\big(Q^h(s_h, s_{h+n}, g), \sum_{i=0}^{n-1} \gamma^i r(s_{h+i}, g) + \gamma^n V^h(s_{h+n}, g)\big)\Big], \tag{3}$$

where $V^h$ is a high-level state value function, $\bar{Q}^h$ is the target network [65], $D$ is a loss function (regression or binary cross-entropy; we use the latter), and the expectations are taken over length-$n$ trajectories $(s_h, a_h, \ldots, s_{h+n})$ and goals $g$ sampled from the dataset. We refer to Appendix E.3 for the full details. We note that one can use any *decoupled* value learning methods (*i.e.*, those that do not involve policy learning, such as IQL [47]) in place of SARSA. For the low-level policy, we can either simply employ goal-conditioned flow BC, or do another round of similar rejection sampling based on a low-level behavioral (SARSA) value function. We call the former variant **SHARSA**[5] and the latter variant **double SHARSA**. We provide the pseudocode for SHARSA and double SHARSA in Algorithms 1 and 2.

SHARSA is appealing for two reasons. First, it is simple and easy to use. SHARSA is only based on behavioral cloning and SARSA, both of which do not require extensive hyperparameter tuning, unlike typical offline RL algorithms [73, 92]. Second, it reduces both the value and policy horizon lengths with an expressive flow policy. This mitigates the curse of horizon in a scalable way.

### 6.3 Results

We now evaluate the performance of various horizon reduction techniques on the main benchmark tasks. We present the data-scaling curves in Figure 9 (see Figure 14 for the training curves). The results show that horizon reduction techniques can indeed unlock the scalability of offline RL on these challenging tasks. We highlight three particularly informative comparisons:

**Value horizon reduction: SAC+BC vs. $n$-step SAC+BC** shows that simply reducing the *value* horizon with $n$-step returns ($n$-step SAC+BC) substantially improves scalability and even *asymptotic* performance on many tasks. This matches our didactic experiments in Section 5.1. We note that their network sizes and training objectives are identical, except for the use of $n$-step returns.

**Policy horizon reduction: Flow BC vs. hierarchical flow BC** shows that reducing the *policy* horizon also significantly improves performance, but on a different set of tasks. In particular, it shows that, on some tasks (*e.g.*, cube-octuple), it is challenging to achieve even non-zero performance without reducing the policy horizon.

---

[5]This acronym stands for (state)–(high-level action)–(reward)–(state)–(high-level action).

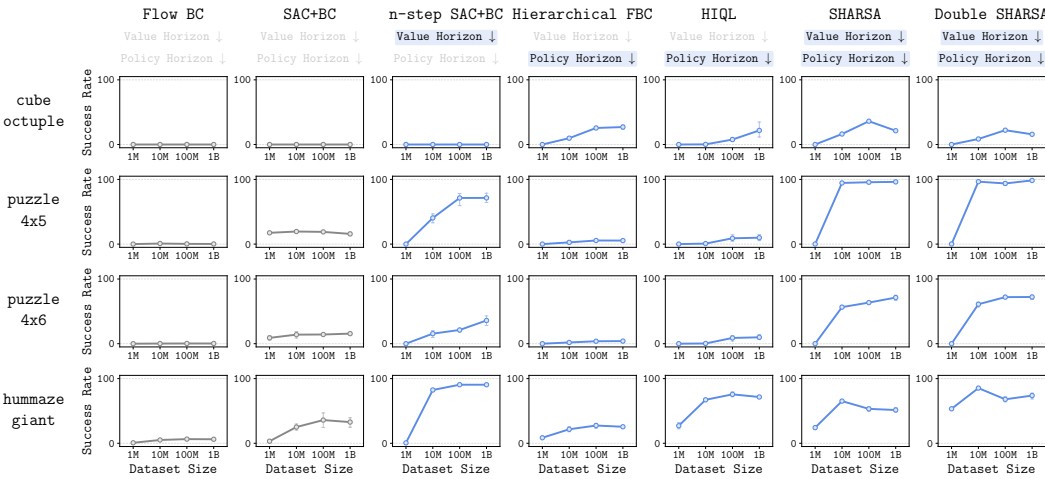

Figure 9: **Horizon reduction makes RL scalable.** Value and policy horizon reduction techniques often result in substantially better scaling and asymptotic performance.

**Value *and* policy horizon reduction: SHARSA vs. the others** shows that reducing *both* value and policy horizons leads to the best of both worlds. In particular, (double) SHARSA is the *only* method that achieves non-trivial performance on all four tasks in our experiments. In Appendix C, we present several ablation studies on SHARSA, discussing the relative importance of various design choices (*e.g.*, alternative policy extraction strategies and value learning objectives).

## 7   Call for research: offline RL algorithms should be evaluated for *scalability*

In this work, we empirically showed that standard, non-hierarchical offline RL methods struggle to scale on complex tasks. We hypothesized that this is due to the curse of horizon, and demonstrated that techniques that explicitly reduce the horizon length, including SHARSA, can unlock scalability.

However, this is far from the end of the story. Empirically, still none of these techniques enable *mastering* all four tasks (*i.e.*, achieving a 100% performance), even with 1B data. Methodologically, these hierarchical methods only *mitigate* the error accumulation issue in TD learning with two-level hierarchies, rather than fundamentally solving it. Moreover, SHARSA and other $n$-step return-based methods implicitly assume that dataset trajectories are near-optimal within short segments (although double SHARSA relaxes this assumption to some extent). Finally, our results still indicate room for improvement over SHARSA, as in some cases the performance does not always scale monotonically with increasing dataset sizes (Figure 9). These limitations of current approaches open up a number of fruitful research questions in scalable reinforcement learning:

- Can we completely avoid TD learning while performing RL (*e.g.*, potentially with model-based RL [29], linear programming [79, 96], or shortest path algorithms [15, 41])?
- Can we find a *simple*, scalable way to extend beyond two-level hierarchies to deal with horizons of arbitrary length?
- Is the curse of horizon fundamentally impossible to solve? The RL theory community suggests otherwise [104, 105], and can we instantiate such a principle within deep RL?

We conclude this paper by calling for research on *scalable* offline RL algorithms, that is, algorithmic research done on large-scale datasets and complex tasks. Currently, offline RL research is often mainly conducted on standard datasets (*e.g.*, D4RL [21], OGBench [75], etc.) with 1M–5M transitions. However, success on small-scale tasks and datasets does not necessarily guarantee success on datasets that are 1000× larger, as not every algorithm *scales* equally [88, 93]. Hence, to assess their potential at scale, it is important to directly evaluate new methods on substantially more challenging tasks and larger datasets and measure scaling trends. To facilitate this, we open-source our tasks, datasets, and implementations (link), where we have made them as easy to use as possible. We hope that our insights in this work, as well as our open-source implementation, serve as a foundation for the development of scalable offline RL objectives that unlock the full potential of data-driven RL.

## Acknowledgments and Disclosure of Funding

We thank Oleh Rybkin for helpful discussions. This work was partly supported by the Korea Foundation for Advanced Studies (KFAS), National Science Foundation Graduate Research Fellowship Program under Grant No. DGE 2146752, ONR N00014-22-1-2773, and NSF IIS-2150826. This research used the Savio computational cluster resource provided by the Berkeley Research Computing program at UC Berkeley.

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

# A  Offline RL scales well on short-horizon tasks

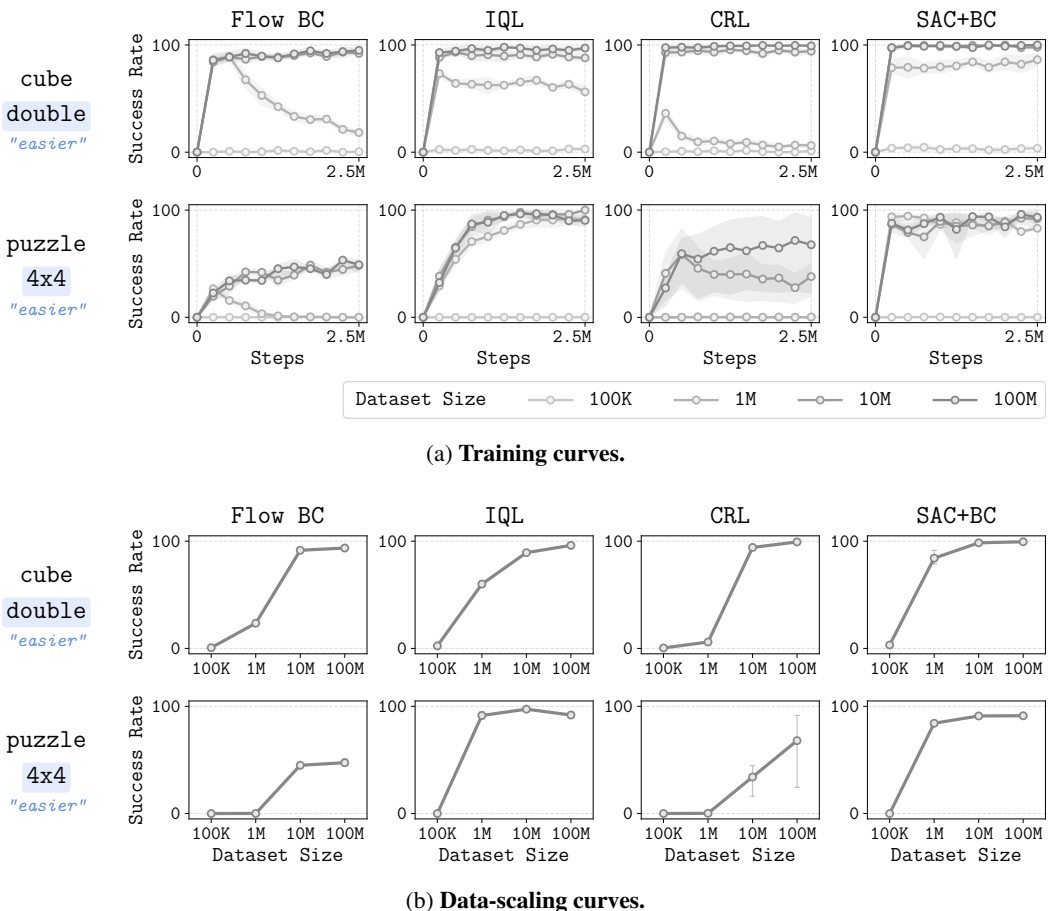

(a) **Training curves.**

(b) **Data-scaling curves.**

Figure 10: **Offline RL scales well on easier, shorter-horizon tasks.** We evaluate flow BC, IQL, CRL, and SAC+BC on the same tasks with fewer objects, and show that they generally scale well on these simpler tasks.

To further verify the validity of our benchmark tasks as well as the offline RL algorithms considered in Section 4, we evaluate these methods on the same tasks with fewer objects: cube-double with 2 cubes (as opposed to cube-octuple with 8 cubes) and puzzle-4x4 with 16 buttons (as opposed to puzzle-4x5 with 20 buttons). Figure 10 shows the training and data-scaling curves of flow BC, IQL, CRL, and SAC+BC on the two tasks. The results suggest that current offline RL algorithms generally scale well on these easier, shorter-horizon tasks. This confirms that our dataset *distribution* provides sufficient coverage to learn a near-optimal policy, and serves as a sanity check for our implementations of these offline RL algorithms

# B  Other attempts to fix the scalability of offline RL

In the main paper, we showed that standard (flat) offline RL methods struggle to scale on complex, long-horizon tasks, and that horizon reduction techniques can effectively address this scalability issue. Are there other solutions to fix scalability without reducing the horizon? We were unable to find any techniques that are as effective as horizon reduction, and we describe failed attempts in this section. Unless otherwise mentioned, we employ SAC+BC and the largest 1B datasets in the experiments below. We note that SAC+BC is the best method in Figure 10, and that behavior-regularized methods of this sort achieve state-of-the-art performance on standard benchmarks [91].

**Larger networks.** The first row of Figure 11 shows the results with larger networks with MLPs and residual MLPs (ResMLPs) [51, 70], up to 591M-sized models. To stabilize training, we reduce the

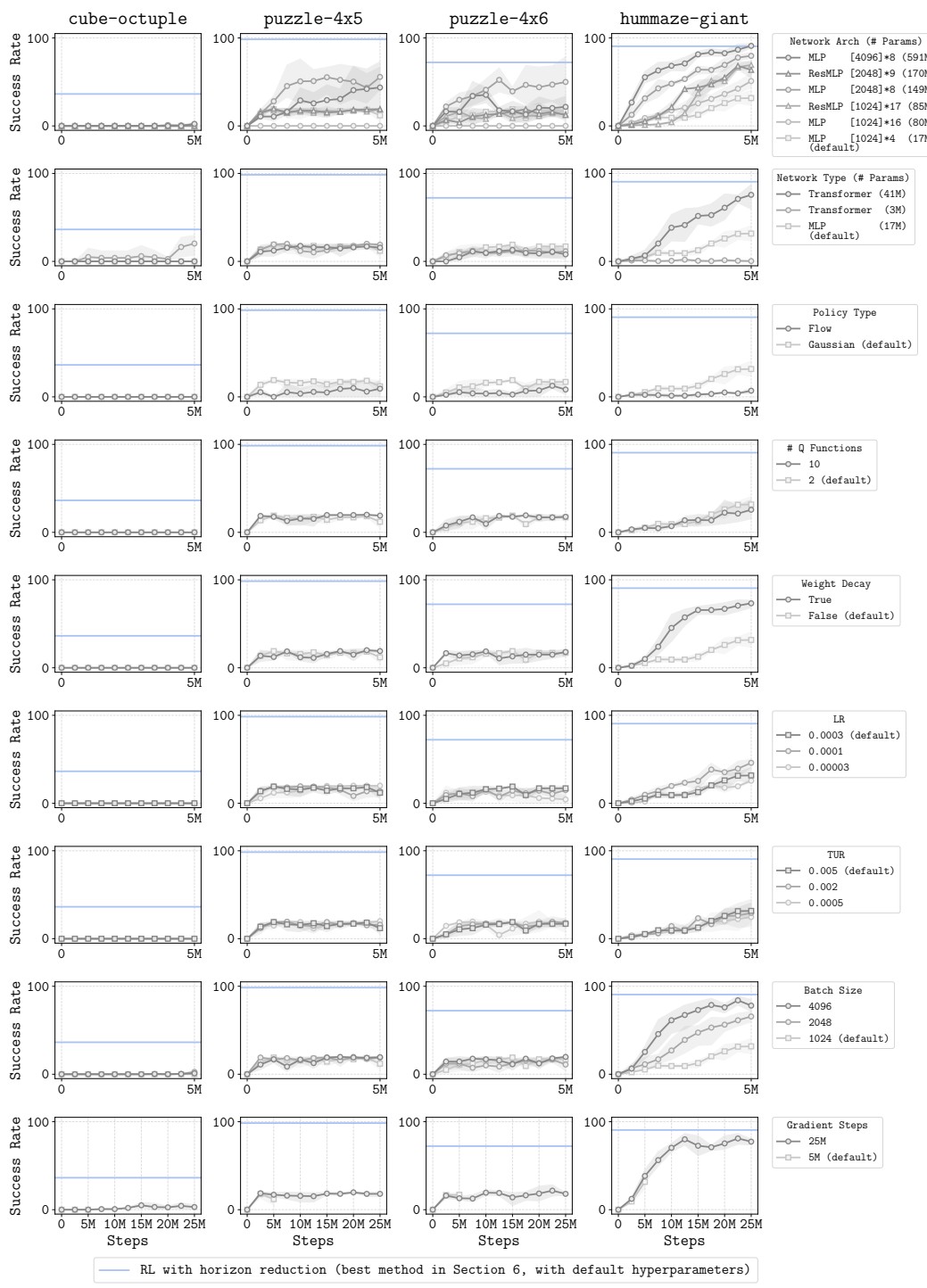

Figure 11: **Nine different attempts to fix the scalability of offline RL *without* horizon reductions.** The results show that **none** of these fixes are as effective as horizon reduction (denoted by the blue line) in general.

learning rate of the largest 591M model from 0.0003 to 0.0001, conceptually following the suggestion by Yang et al. [101]. The results suggest that while larger networks can improve performance to some degree, simply increasing the capacity is not sufficient to master the tasks. On the other hand, horizon reduction enables significantly better asymptotic performance (denoted in blue) even with the default-sized models. We refer to the main paper (Section 4) for further discussion.

**Transformers.** We investigate whether replacing MLPs with Transformers [95] can improve performance. To handle vector-valued inputs with a Transformer, we first map the input to a $T_r$-dimensional vector using a dense layer, reshape it into a length-$T_\ell$ sequence of $T_k$-dimensional vectors, pass it through $T_n$ self-attention blocks (with $T_m$ MLP units) with four independent heads, and concatenate the outputs for the final dense layer. We employ Transformers of two different sizes with $(T_r, T_\ell, T_k, T_n, T_m) = (2048, 16, 128, 4, 128)$ and $(2048, 8, 256, 10, 1024)$. The former network has 3M total parameters and the latter has 41M total parameters. Due to the significantly higher computational cost, we use a smaller batch size (256 instead of 1024) for runs with the larger Transformer, so that each run completes within three days. The second row of Figure 11 shows the results with Transformers. These results suggest that while using Transformers improves performance on some tasks, it still often falls significantly short of horizon reduction techniques.

**More expressive policies.** To understand whether a more expressive policy can improve performance, we train (goal-conditioned) FQL [76], one of the closest methods to SAC+BC that use expressive flow policies [2, 59, 61]. The third row of Figure 11 presents the results, which suggest that simply changing the policy class does not improve performance on the four benchmark tasks.

**Larger Q ensembles.** The fourth row of Figure 11 compares the results with 2 (default) and 10 Q networks. The results show that their performances are nearly identical.

**Regularization.** To understand whether additional regularization can address the scalability issue, we evaluate performance with weight decay (with a coefficient of $0.01$, selected from $\{0.0001, 0.001, 0.01, 0.1\}$). We note that we use layer normalization [4] by default for all networks. The fifth row of Figure 11 shows the results. While weight decay yields a non-trivial improvement on one task (`humanoidmaze-giant`), it does not improve performance on the other three, more challenging tasks.

**Smaller learning rates (LRs) and target network update rates (TURs).** The sixth and seventh rows of Figure 11 show the results with different learning rates and target network update rates. These results indicate that simply adjusting these hyperparameters does not substantially improve performance on the benchmark tasks.

**Larger batch sizes.** The eighth row of Figure 11 shows the results with larger batch sizes. While larger batches help on `humanoidmaze-giant`, they do not improve performance on the other three tasks.

**Longer training.** The ninth row of Figure 11 shows the results with $5\times$ longer training (25M gradient steps in total). While extended training improves performance on `humanoidmaze-giant`, it does not yield significant improvements on the other three tasks.

**Other attempts.** In the earlier stages of this research, we tried a classification-based loss with HL-Gauss [20, 37], but it did not lead to a significant improvement in performance. We also tried residual TD error minimization [83] (*i.e.*, removing the stop-gradient in the TD target), but we were unable to achieve non-trivial performance with the residual loss.

## C   Ablation studies of SHARSA

In this section, we present three ablation studies on the design choices of SHARSA. All results are evaluated on the largest 1B datasets.

**Value learning methods.** While SHARSA uses SARSA for the value learning algorithm, we can in principle use any decoupled value learning algorithm (*i.e.*, one that does not involve policy learning) in place of SARSA, such as IQL [47] or its variants [25, 100]. The first row of Figure 12 compares the performance of three different value learning methods within the SHARSA framework: SARSA, IQL with $\kappa = 0.7$, and IQL with $\kappa = 0.9$, where $\kappa$ is the expectile hyperparameter in IQL (Appendix E.1). The results suggest that the simplest SARSA algorithm is sufficient to achieve the best performance on our benchmark tasks, which partly aligns with recent findings [7, 18, 50].

**Policy extraction methods.** SHARSA uses rejection sampling for high-level policy extraction. In the main paper, we discussed how reparameterized gradient-based approaches may not be suitable for *high-level* policy extraction, due to potentially ill-defined first-order gradient information in the state space. To empirically confirm this, we replace rejection sampling in SHARSA with two alternative policy extraction methods based on reparameterized gradients: DDPG+BC [22, 73] and FQL [76].

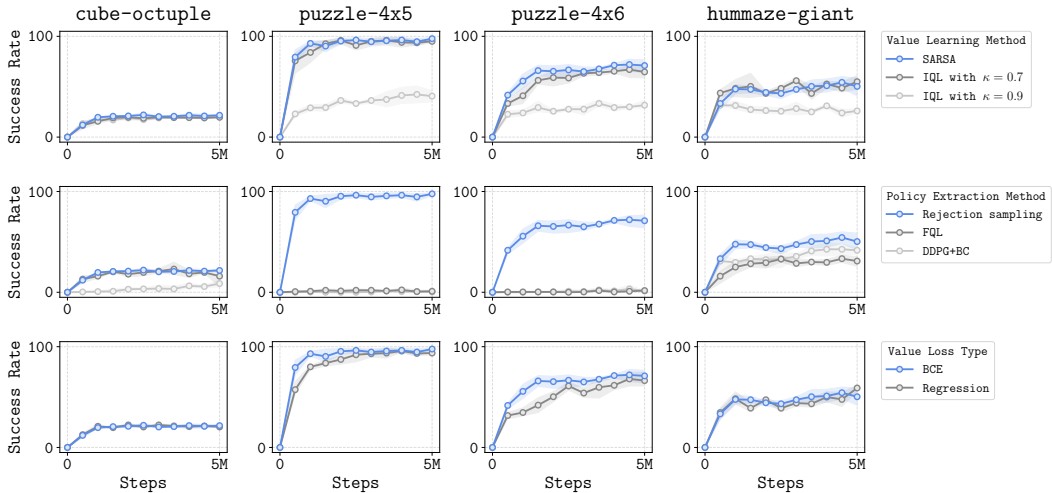

Figure 12: **Ablation studies of SHARSA.**

The former extracts a (high-level) Gaussian policy and the latter extracts a (high-level) flow policy. We recall that SHARSA uses goal-conditioned BC for the low-level policy. The second row of Figure 12 presents the results. As expected, the results show that these reparameterized gradient-based methods perform worse than rejection sampling, especially on the `puzzle` tasks, which contain discrete information (*e.g.*, button states) in the state space.

**Value losses.** As explained in Appendix E.3, we employ the binary cross-entropy (BCE) loss (instead of the more commonly used regression loss) for the value losses in SHARSA (Equations (25) and (26)). The third row of Figure 12 compares these two choices, showing that the BCE loss leads to better performance and faster convergence. While we do not provide separate plots, we found that the BCE loss generally results in better performance, regardless of the underlying algorithms.

# D   Additional results

Table 1: **Horizon reduction improves performance in reward-based (non-goal-conditioned) RL too.**

| Task | SARSA | IQL | SAC+BC | n-SAC+BC | SHARSA ($\kappa = 0.5$) | SHARSA ($\kappa = 0.7$) |
|---|---|---|---|---|---|---|
| Horizon Reduction Type | - | - | - | Value | Value & policy | Value & Policy |
| cube-quadruple-play-singletask-task1-v0 | $0_{\pm0}$ | $7_{\pm6}$ | $0_{\pm0}$ | $0_{\pm0}$ | $50_{\pm9}$ | $\mathbf{60}_{\pm8}$ |
| puzzle-4x5-play-singletask-task1-v0 | $4_{\pm3}$ | $24_{\pm7}$ | $0_{\pm0}$ | $0_{\pm0}$ | $\mathbf{94}_{\pm3}$ | $\mathbf{94}_{\pm2}$ |
| puzzle-4x6-play-singletask-task1-v0 | $2_{\pm2}$ | $5_{\pm2}$ | $0_{\pm0}$ | $0_{\pm0}$ | $\mathbf{14}_{\pm6}$ | $\mathbf{14}_{\pm7}$ |
| humanoidmaze-giant-navigate-singletask-task1-v0 | $0_{\pm0}$ | $2_{\pm2}$ | $44_{\pm35}$ | $\mathbf{88}_{\pm3}$ | $26_{\pm4}$ | $\mathbf{87}_{\pm3}$ |

**Results on reward-based tasks.** While we focus on goal-conditioned tasks in this work, the benefits of horizon reduction are not limited to goal-conditioned RL. To empirically demonstrate this, we additionally evaluate two horizon reduction techniques, $n$-step SAC+BC (which reduces the value horizon) and SHARSA (which reduces both the value and policy horizons), on four reward-based `singletask` tasks from OGBench [75]. We employ 100M-sized (`cube`) and 1B-sized (others) datasets.

On these tasks, we evaluate SARSA, IQL (with $\kappa = 0.7$), SAC+BC, $n$-step SAC+BC, and SHARSA. Additionally, we consider an IQL variant of SHARSA (with $\kappa = 0.7$, Appendix C), which can be helpful as these `singletask` tasks have higher suboptimality due to the absence of hindsight relabeling. We use AWR with $\alpha = 10$ for SARSA and IQL, and BC regularization with $\alpha = 0.01$ (`humanoidmaze`) or $0.1$ (others) for SAC+BC and $n$-step SAC+BC.

Table 1 shows the performance measured at the 1M epoch. The results suggest that these horizon reduction techniques significantly improve performance in reward-based offline RL as well.

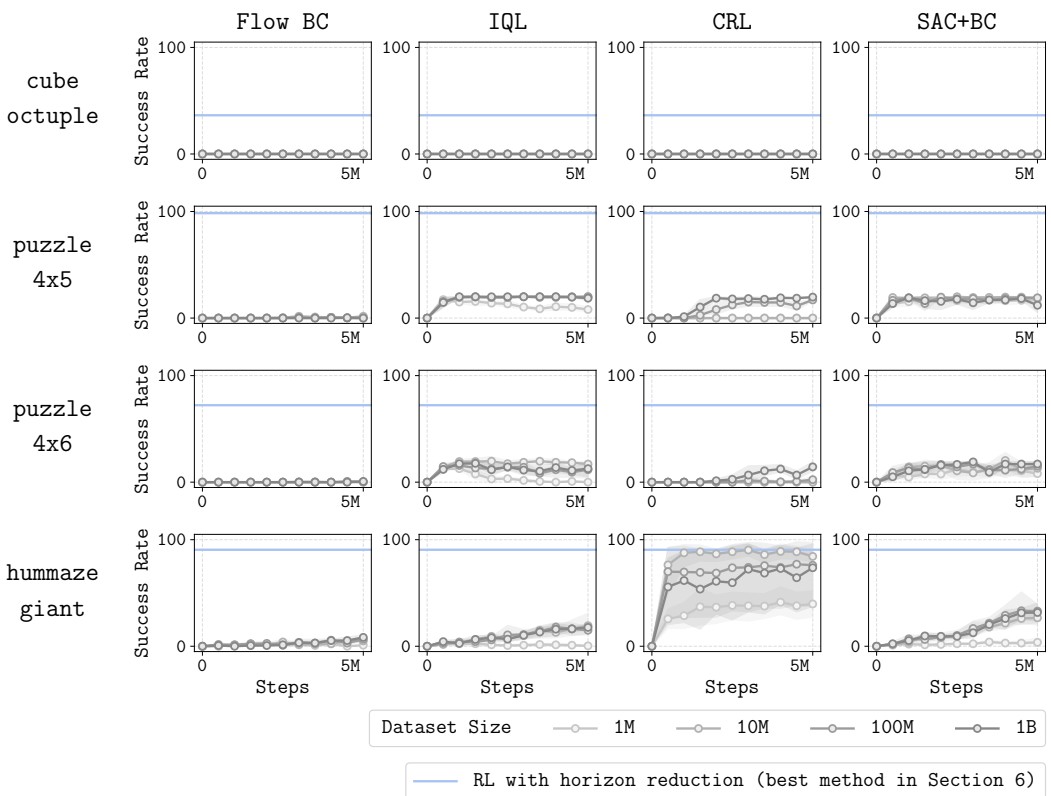

Figure 13: **Training curves of standard offline RL methods.**

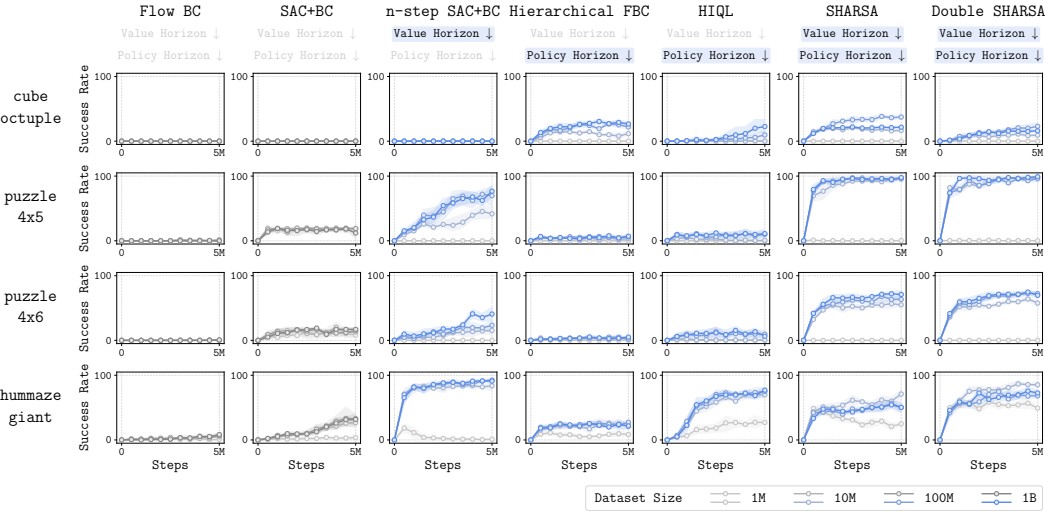

Figure 14: **Training curves of horizon reduction techniques.**

**Full training curves.** Figures 13 and 14 provide the full training curves of the methods considered in Figures 3 and 9, respectively.

# E    Offline RL algorithms

In this section, we describe the offline (goal-conditioned) RL algorithms considered in this work. In the below, $\gamma \in [0, 1]$ denotes the discount factor and $\mathcal{G}$ denotes the goal space, which is the domain of a goal specification function $\varphi_g(s) : \mathcal{S} \to \mathcal{G}$. For example, in `humanoidmaze`, $\varphi_g$ is a function that outputs only the $x$-$y$ coordinates of the state. We also assume that the action space is a $d$-dimensional Euclidean space (*i.e.*, $\mathcal{A} = \mathbb{R}^d$), unless otherwise mentioned. We denote network parameters as $\theta$ (with corresponding subscripts when there are multiple networks). The goal-conditioned reward function $r(s, g) : \mathcal{S} \times \mathcal{G} \to \mathbb{R}$ is given as either $[s = g]$ (the 0-1 sparse reward function) or $[s = g] - 1$ (the $-1$-0 sparse reward function), where $[\cdot]$ is the Iverson bracket (*i.e.*, the indicator function for propositions). We use the former for classification-based methods and the latter for regression-based methods.

We denote the state-action-goal sampling distribution as $p^{\mathcal{D}}$. In general, $p^{\mathcal{D}}(s, a)$ is the uniform distribution over the dataset state-action pairs and $p^{\mathcal{D}}(g \mid s, a)$ is a mixture of the four distributions: the Dirac delta distribution at the current state ($p^{\mathcal{D}}_{\text{cur}}$), a geometric distribution over the future states ($p^{\mathcal{D}}_{\text{geom}}$), the uniform distribution over the future states ($p^{\mathcal{D}}_{\text{traj}}$), and the uniform distribution over the dataset states ($p^{\mathcal{D}}_{\text{rand}}$). We refer to Park et al. [75] for the full details. The ratios of these four distributions are tunable hyperparameters, which we specify in Table 3.

## E.1    Flat offline RL algorithms

**Flow behavioral cloning (flow BC) [6, 12, 76].** Goal-conditioned flow behavioral cloning trains a vector field $v_\theta(t, s, z, g) : [0, 1] \times \mathcal{S} \times \mathbb{R}^d \times \mathcal{G} \to \mathbb{R}^d$ that generates behavioral action distributions. It minimizes the following objective:

$$L(\theta) = \mathbb{E}_{\substack{s,a \sim p^{\mathcal{D}}(s,a),\ g \sim p^{\mathcal{D}}(g|s,a), \\ z \sim \mathcal{N}(0, I_d),\ t \sim \text{Unif}([0,1]), \\ a^t = (1-t)z + ta}} \left[ \left\| v_\theta(t, s, a^t, g) - (a - z) \right\|_2^2 \right], \tag{4}$$

where $\text{Unif}([0, 1])$ denotes the uniform distribution over the interval $[0, 1]$.

After training the vector field $v_\theta$, actions are obtained by solving the ordinary differential equation (ODE) corresponding to the *flow* [52] generated by the vector field. We use the Euler method with a step count of 10, following prior work [76]. See Lipman et al. [60], Park et al. [76] for further discussions about flow matching and flow policies.

**Implicit Q-learning (IQL) [47, 72].** Goal-conditioned IQL trains a state value function $V_{\theta_V}(s, g) : \mathcal{S} \times \mathcal{G} \to \mathbb{R}$ and a state-action value function $Q_{\theta_Q}(s, a, g) : \mathcal{S} \times \mathcal{A} \times \mathcal{G} \to \mathbb{R}$ with the following losses:

$$L^V(\theta_V) = \mathbb{E}_{(s,a) \sim p^{\mathcal{D}}(s,a),\ g \sim p^{\mathcal{D}}(g|s,a)} \left[ \ell_\kappa^2 \left( V_{\theta_V}(s, g) - Q_{\bar{\theta}_Q}(s, a, g) \right) \right], \tag{5}$$

$$L^Q(\theta_Q) = \mathbb{E}_{(s,a,s') \sim p^{\mathcal{D}}(s,a,s'),\ g \sim p^{\mathcal{D}}(g|s,a)} \left[ \left( Q_{\theta_Q}(s, a, g) - r(s, g) - \gamma V_{\theta_V}(s', g) \right)^2 \right], \tag{6}$$

where $\ell_\kappa^2$ denotes the expectile loss, $\ell_\kappa^2(x) = |\kappa - [x < 0]| x^2$, and $\bar{\theta}_Q$ denotes the parameters of the target Q network [65].

From the learned Q function, it extracts a (Gaussian) policy $\pi_{\theta_\pi}(a \mid s, g) : \mathcal{S} \times \mathcal{G} \to \Delta(\mathcal{A})$ by maximizing the following DDPG+BC objective [73]:

$$J^\pi(\theta_\pi) = \mathbb{E}_{(s,a) \sim p^{\mathcal{D}}(s,a),\ g \sim p^{\mathcal{D}}(g|s,a)} \left[ Q_{\theta_Q}(s, \mu_{\theta_\pi}(s, g), g) + \alpha \log \pi_{\theta_\pi}(a \mid s, g) \right], \tag{7}$$

where $\mu_{\theta_\pi}$ denotes the mean of the Gaussian policy $\pi_{\theta_\pi}$ and $\alpha$ denotes the hyperparameter that controls the strength of the BC regularizer. While the original IQL method uses the AWR objective [77], we use DDPG+BC as Park et al. [73] found it to scale better than AWR.

**Contrastive reinforcement learning (CRL) [18].** CRL trains a logarithmic goal-conditioned value function $f_{\theta_f}(s, a, g) : \mathcal{S} \times \mathcal{A} \times \mathcal{G} \to \mathbb{R}$ with the following binary noise contrastive estimation objective [63]:

$$J^f(\theta_f) = \mathbb{E}_{\substack{(s,a) \sim p^{\mathcal{D}}(s,a), g \sim p^{\mathcal{D}}_+(g|s,a), \\ g^- \sim p^{\mathcal{D}}_-(g)}} \left[ \log \sigma(f_{\theta_f}(s, a, g)) + \log(1 - \sigma(f_{\theta_f}(s, a, g^-))) \right], \tag{8}$$

where $\sigma\colon \mathbb{R} \to (0,1)$ is the sigmoid function, $p_+^{\mathcal{D}}$ is a geometric future goal sampling distribution, and $p_-^{\mathcal{D}}$ is the uniform goal sampling distribution. Eysenbach et al. [18] show that the optimal solution $f^*$ to the above objective is given by $f^*(s,a,g) = \log Q^{\mathrm{MC}}(s,a,g) + C(g)$, where $Q^{\mathrm{MC}}$ is the Monte-Carlo value function and $C$ is a function that does not depend on $s$ and $a$. As in Eysenbach et al. [18], we employ an inner product parameterization to model $f_{\theta_f}$ as $f(s,a,g) = \psi_1(s,a)^\top \psi_2(g)$ (where we omit the parameter dependencies for simplicity) with $k$-dimensional representations, $\psi_1 : \mathcal{S} \times \mathcal{A} \to \mathbb{R}^k$ and $\psi_2 : \mathcal{G} \to \mathbb{R}^k$.

Similarly to IQL, CRL extracts a policy with the following DDPG+BC objective:

$$J^\pi(\theta_\pi) = \mathbb{E}_{(s,a)\sim p^{\mathcal{D}}(s,a),\, g\sim p^{\mathcal{D}}(g|s,a)}\left[f_{\theta_f}(s, \mu_{\theta_\pi}(s,g), g) + \alpha \log \pi_{\theta_\pi}(a \mid s,g)\right]. \qquad (9)$$

**Soft actor-critic + behavioral cloning (SAC+BC).** SAC+BC is the SAC [27] variant of TD3+BC [22, 91]. We found SAC+BC to be generally better than both TD3+BC [22] and its successor ReBRAC [91] due to the use of stochastic actions in the actor loss (note that TD3 [23] uses deterministic actions in the actor objective), which serves as a regularizer in the offline RL setting. Goal-conditioned SAC+BC minimizes $L^Q$ and maximizes $J^\pi$ below to train a Q function $Q_{\theta_Q}$ and a policy $\pi_{\theta_\pi}$:

$$L^Q(\theta_Q) = \mathbb{E}_{\substack{(s,a,s')\sim p^{\mathcal{D}}(s,a,s'),\, g\sim p^{\mathcal{D}}(g|s,a),\\ a^\pi \sim \pi_{\theta_\pi}(a|s',g)}}\left[\left(Q_{\theta_Q}(s,a,g) - r(s,g) - \gamma Q_{\bar{\theta}_Q}(s',a^\pi,g)\right)^2\right], \qquad (10)$$

$$J^\pi(\theta_\pi) = \mathbb{E}_{\substack{(s,a)\sim p^{\mathcal{D}}(s,a),\\ a^\pi \sim \pi_{\theta_\pi}(a|s,g)}}\left[Q_{\theta_Q}(s,a^\pi,g) - \alpha\|a^\pi - a\|_2^2 - \lambda \log \pi_{\theta_\pi}(a^\pi \mid s,g)\right], \qquad (11)$$

where $\alpha$ is the hyperparameter for the BC strength and $\lambda$ is the entropy regularizer (which is often automatically adjusted with dual gradient descent to match a target entropy value [28]).

On goal-conditioned tasks with 0-1 sparse rewards, since we know that the optimal Q values are always in between 0 and 1, we can instead use the following *binary cross-entropy (BCE)* variant for the Q loss [42]:

$$L_{\mathrm{BCE}}^Q(\theta_Q) = \mathbb{E}_{\substack{(s,a,s')\sim p^{\mathcal{D}}(s,a,s'),\, g\sim p^{\mathcal{D}}(g|s,a),\\ a^\pi \sim \pi_{\theta_\pi}(a|s',g)}}\left[\mathrm{BCE}\left(Q_{\theta_Q}(s,a,g), r(s,g) + \gamma Q_{\bar{\theta}_Q}(s',a^\pi,g)\right)\right], \tag{12}$$

where $\mathrm{BCE}(x,y) = -y\log x - (1-y)\log(1-x)$. We found this variant to be generally better than the original regression objective, as it focuses better on small differences in low Q values, which is crucial for extracting policies on long-horizon tasks. When using the binary cross-entropy variant, we model the *logits* of Q values (instead of the raw Q values) with a neural network, and use the logit values in place of the Q values in the SAC+BC actor objective as follows:

$$J_{\mathrm{BCE}}^\pi(\theta_\pi) = \mathbb{E}_{\substack{(s,a)\sim p^{\mathcal{D}}(s,a),\\ a^\pi \sim \pi_{\theta_\pi}(a|s,g)}}\left[\mathrm{logit}\, Q_{\theta_Q}(s,a^\pi,g) - \alpha\|a^\pi - a\|_2^2 - \lambda \log \pi_{\theta_\pi}(a^\pi \mid s,g)\right]. \qquad (13)$$

We found the use of logits in the actor loss to be crucial on long-horizon tasks, as it applies more uniform behavioral constraints across the state space.

**Flow Q-learning (FQL) [76].** FQL is a behavior-regularized offline RL algorithm that trains a flow policy with one-step distillation. Goal-conditioned FQL trains a Q function $Q_{\theta_Q}(s,a,g)$ : $\mathcal{S} \times \mathcal{A} \times \mathcal{G} \to \mathbb{R}$, a BC vector field $v_{\theta_\pi}(t,s,z,g) : [0,1] \times \mathcal{S} \times \mathbb{R}^d \times \mathcal{G} \to \mathbb{R}^d$ that generates a noise-conditioned flow BC policy $\mu_{\theta_\pi}(s,z,g) : \mathcal{S} \times \mathbb{R}^d \times \mathcal{G} \to \mathcal{A}$, and a noise-conditioned one-step policy $\mu_{\theta_\mu}(s,z,g) : \mathcal{S} \times \mathbb{R}^d \times \mathcal{G} \to \mathcal{A}$, with the following losses:

$$L^\pi(\theta_\pi) = \mathbb{E}_{\substack{(s,a)\sim p^{\mathcal{D}}(s,a),\, g\sim p^{\mathcal{D}}(g|s,a),\\ z\sim \mathcal{N}(0,I_d),\, t\sim \mathrm{Unif}([0,1]),\\ a^t = (1-t)z + ta}}\left[\left\|v_{\theta_\pi}(t,s,a^t,g) - (a-z)\right\|_2^2\right], \qquad (14)$$

$$L^Q(\theta_Q) = \mathbb{E}_{\substack{(s,a,s')\sim p^{\mathcal{D}}(s,a,s'),\, g\sim p^{\mathcal{D}}(g|s,a),\\ z\sim \mathcal{N}(0,I_d),\, a^\pi = \mu_{\theta_\mu}(s',z,g)}}\left[\left(Q_{\theta_Q}(s,a,g) - r(s,g) - \gamma Q_{\bar{\theta}_Q}(s',a^\pi,g)\right)^2\right], \qquad (15)$$

$$L^\mu(\theta_\mu) = \mathbb{E}_{\substack{(s,a)\sim p^{\mathcal{D}}(s,a),\\ g\sim p^{\mathcal{D}}(g|s,a),\\ z\sim \mathcal{N}(0,I_d)}}\left[-Q(s,\mu_{\theta_\mu}(s,z,g),g) + \alpha\|\mu_{\theta_\mu}(s,z,g) - \mu_{\theta_\pi}(s,z,g)\|_2^2\right], \qquad (16)$$

where $\alpha$ is the BC coefficient. The output of FQL is the one-step policy $\mu_{\theta_\mu}$. In our experiments, we use the binary cross-entropy variant of FQL in our experiments, which replaces the regression loss in Equation (15) with the corresponding binary cross-entropy loss, as in Equation (12).

## E.2 Hierarchical offline RL algorithms

**$n$-step soft actor-critic + behavioral cloning ($n$-step SAC+BC).** $n$-step SAC+BC is a variant of SAC+BC that employs $n$-step returns. The only difference from SAC+BC is that it minimizes the following value loss:

$$L^Q(\theta_Q) = \mathbb{E}_{\substack{(s_h, a_h, \ldots, s_{h+n}) \sim p^{\mathcal{D}}, \\ g \sim p^{\mathcal{D}}(g \mid s_h, a_h), \\ a^\pi \sim \pi_{\theta_\pi}(a \mid s_{h+n}, g)}} \left[ D\left( Q_{\theta_Q}(s_h, a_h, g), \sum_{i=0}^{n-1} \gamma^i r(s_{h+i}, g) + \gamma^n Q_{\bar\theta_Q}(s_{h+n}, a^\pi, g) \right) \right],$$

(17)

where we omit the arguments in $p^{\mathcal{D}}(s_h, a_h, \ldots, s_{h+n})$ and $D$ is either the regression loss $\mathrm{Reg}(x, y) = (x - y)^2$ or the binary cross-entropy loss $\mathrm{BCE}(x, y) = -y \log x - (1 - y) \log(1 - x)$. We found that the BCE loss performs and scales better in our experiments. In practice, we also need to handle several edge cases involving truncated trajectories and goals in the above loss; we refer the reader to our implementation for further details.

**Hierarchical flow BC (hierarchical FBC).** Hierarchical flow BC trains two policies: a high-level policy $\pi_{\theta_h}^h(w \mid s, g) : \mathcal{S} \times \mathcal{G} \to \Delta(\mathcal{G})$ and a low-level policy $\pi_{\theta_\ell}^\ell(a \mid s, w) : \mathcal{S} \times \mathcal{G} \to \Delta(\mathcal{A})$, where we denote subgoals by $w$. The high-level policy is trained to predict subgoals that are $n$ steps away from the current state, and the low-level policy is trained to predict actions to reach the given subgoal. Both policies are modeled by flows, with vector fields $v_{\theta_h}^h(t, s, z, g) : [0, 1] \times \mathcal{S} \times \mathbb{R}^m \times \mathcal{G} \to \mathbb{R}^m$ and $v_{\theta_\ell}^\ell(t, s, z, w) : [0, 1] \times \mathcal{S} \times \mathbb{R}^d \times \mathcal{G} \to \mathbb{R}^d$, where we assume that the goal space is $\mathcal{G} = \mathbb{R}^m$. These vector fields are trained with the following flow-matching losses:

$$L^h(\theta_h) = \mathbb{E}_{\substack{(s_h, a_h, \ldots, s_{h+n}) \sim p^{\mathcal{D}}, \ g \sim p^{\mathcal{D}}(g \mid s_h, a_h), \\ z \sim \mathcal{N}(0, I_m), \ t \sim \mathrm{Unif}([0, 1]), \\ w^t = (1-t)z + t\varphi_g(s_{t+h})}} \left[ \left\| v_{\theta_h}^h(t, s_h, w^t, g) - (\varphi_g(s_{t+h}) - z) \right\|_2^2 \right], \quad (18)$$

$$L^\ell(\theta_\ell) = \mathbb{E}_{\substack{(s_h, a_h, \ldots, s_{h+n}) \sim p^{\mathcal{D}}, \ z \sim \mathcal{N}(0, I_d), \\ t \sim \mathrm{Unif}([0, 1]), \ a^t = (1-t)z + ta_h}} \left[ \left\| v_{\theta_\ell}^\ell(t, s_h, a^t, s_{h+n}) - (a_h - z) \right\|_2^2 \right]. \quad (19)$$

**Hierarchical implicit Q-learning (HIQL) [72].** HIQL trains a single goal-conditioned value function with implicit V-learning (IVL) [75], and extract hierarchical policies ($\pi_{\theta_h}^h$ and $\pi_{\theta_\ell}^\ell$) with AWR-like objectives [77]. It trains a value function $V_{\theta_V}(s, g) : \mathcal{S} \times \mathcal{G} \to \mathbb{R}$ that is parameterized as $V_{\theta_V}(s, g) = \tilde{V}_{\theta_V}(s, \psi_{\theta_V}(s, g))$ with a representation function $\psi_{\theta_V}(s, g) : \mathcal{S} \times \mathcal{G} \to \mathbb{R}^k$ and a remainder network $\tilde{V}_{\theta_V}(s, z) : \mathcal{S} \times \mathbb{R}^k \to \mathbb{R}$, where we do not distinguish the parameters for $\psi, \tilde{V}$, and $V$ to emphasize that they are part of the value network. The IVL value loss is as follows:

$$L^V(\theta_V) = \mathbb{E}_{(s, a, s') \sim p^{\mathcal{D}}(s, a, s'), \ g \sim p^{\mathcal{D}}(g \mid s, a)} \left[ \ell_\kappa^2 \left( V_{\theta_V}(s, g) - r(s, g) - \gamma V_{\bar\theta_V}(s', g) \right) \right], \quad (20)$$

where $\ell_\kappa^2$ is the expectile loss described in Appendix E.1. From the value function, it extracts two policies by maximizing the following AWR objectives:

$$J^h(\theta_h) = \mathbb{E}_{\substack{(s_h, a_h, \ldots, s_{h+n}) \sim p^{\mathcal{D}}, \\ g \sim p^{\mathcal{D}}(g \mid s_h, a_h)}} \left[ e^{\alpha(V(s_{h+n}, g) - V(s_h, g))} \log \pi_{\theta_h}^h(\psi_{\theta_V}(s_h, s_{h+n}) \mid s_h, g) \right], \quad (21)$$

$$J^\ell(\theta_\ell) = \mathbb{E}_{(s_h, a_h, \ldots, s_{h+n}) \sim p^{\mathcal{D}}} \left[ e^{\alpha(V(s_{h+1}, s_{h+n}) - V(s_h, s_{h+n}))} \log \pi_{\theta_\ell}^\ell(a_h \mid s_h, \psi_{\theta_V}(s_h, s_{h+n})) \right],$$

(22)

where $\alpha$ is the inverse temperature hyperparameter for AWR. Similar to $n$-step SAC+BC, there are several edge cases with truncated trajectories and goals, and we refer to our implementation for the full details.

## E.3 SHARSA

**SHARSA.** SHARSA is our newly proposed offline RL algorithm based on hierarchical flow BC and $n$-step SARSA. It has the following components:

- High-level BC flow policy $\pi_{\beta, \theta_h}^h(w \mid s, g) : \mathcal{S} \times \mathcal{G} \to \Delta(\mathcal{G})$,
- Low-level BC flow policy $\pi_{\beta, \theta_\ell}^\ell(a \mid s, w) : \mathcal{S} \times \mathcal{G} \to \Delta(\mathcal{A})$,

- $n$-step Q function $Q_{\theta_Q}(s, w, g) : \mathcal{S} \times \mathcal{G} \times \mathcal{G} \to \mathbb{R}$,
- $n$-step V function $V_{\theta_V}(s, g) : \mathcal{S} \times \mathcal{G} \to \mathbb{R}$.

As in hierarchical FBC, the policies are modeled by vector fields, $v^h_{\theta_h}(t, s, z, g) : [0, 1] \times \mathcal{S} \times \mathbb{R}^m \times \mathcal{G} \to \mathbb{R}^m$ and $v^\ell_{\theta_\ell}(t, s, z, w) : [0, 1] \times \mathcal{S} \times \mathbb{R}^d \times \mathcal{G} \to \mathbb{R}^d$, where we recall that $\mathcal{G} = \mathbb{R}^m$ and $\mathcal{A} = \mathbb{R}^d$. They are trained via the following flow behavioral cloning losses:

$$L^h(\theta_h) = \mathbb{E}_{\substack{(s_h, a_h, \ldots, s_{h+n}) \sim p^{\mathcal{D}},\ g \sim p^{\mathcal{D}}(g | s_h, a_h), \\ z \sim \mathcal{N}(0, I_m),\ t \sim \mathrm{Unif}([0,1]), \\ w^t = (1-t)z + t\varphi_g(s_{t+h})}} \left[ \left\| v^h_{\theta_h}(t, s_h, w^t, g) - (\varphi_g(s_{t+h}) - z) \right\|_2^2 \right], \quad (23)$$

$$L^\ell(\theta_\ell) = \mathbb{E}_{\substack{(s_h, a_h, \ldots, s_{h+n}) \sim p^{\mathcal{D}},\ z \sim \mathcal{N}(0, I_d), \\ t \sim \mathrm{Unif}([0,1]),\ a^t = (1-t)z + ta_h}} \left[ \left\| v^\ell_{\theta_\ell}(t, s_h, a^t, s_{h+n}) - (a_h - z) \right\|_2^2 \right], \quad (24)$$

where we recall that $\varphi_g$ is the goal specification function defined in the first paragraph of Appendix E. The value functions are trained with the following SARSA losses:

$$L^V(\theta_V) = \mathbb{E}_{\substack{(s_h, a_h, \ldots, s_{h+n}) \sim p^{\mathcal{D}}, \\ g \sim p^{\mathcal{D}}(g | s_h, a_h)}} \left[ D\left( V^h_{\theta_V}(s_h, g), Q^h_{\bar{\theta}_Q}(s_h, s_{h+n}, g) \right) \right], \quad (25)$$

$$L^Q(\theta_Q) = \mathbb{E}_{\substack{(s_h, a_h, \ldots, s_{h+n}) \sim p^{\mathcal{D}}, \\ g \sim p^{\mathcal{D}}(g | s_h, a_h)}} \left[ D\left( Q^h_{\theta_Q}(s_h, s_{h+n}, g), \sum_{i=0}^{n-1} \gamma^i r(s_{h+i}, g) + \gamma^n V^h_{\theta_V}(s_{h+n}, g) \right) \right], \quad (26)$$

where $D$ is either the regression loss $\mathrm{Reg}(x, y) = (x - y)^2$ or the binary cross-entropy loss $\mathrm{BCE}(x, y) = -y \log x - (1 - y) \log(1 - x)$. As before, we found that the BCE variant works better on long-horizon tasks.

At test time, we employ rejection sampling for the high-level policy. Specifically, it defines the distribution of the high-level policy $\pi^h_{\theta_h}(w \mid s, g) : \mathcal{S} \times \mathcal{G} \to \Delta(\mathcal{G})$ as follows:

$$\pi^h_{\theta_h}(s, g) \stackrel{d}{=} \underset{w_1, \ldots, w_N : w_i \sim \pi^h_{\beta, \theta_h}(w | s, g)}{\arg\max} Q^h_{\theta_Q}(s, w_i, g), \quad (27)$$

where $N$ is the number of samples. SHARSA simply uses the behavioral low-level policy; *i.e.*, $\pi^\ell_{\theta_\ell}(s, w) \stackrel{d}{=} \pi^\ell_{\beta, \theta_\ell}(s, w)$. We provide the pseudocode in Algorithm 1.

**Double SHARSA.** Double SHARSA employs an additional round of rejection sampling in the low-level policy to further enhance optimality. To do this, it defines additional low-level value networks:

- Low-level Q function $Q^\ell_{\theta_q}(s, a, w) : \mathcal{S} \times \mathcal{A} \times \mathcal{G} \to \mathbb{R}$,
- Low-level V function $V^\ell_{\theta_v}(s, w) : \mathcal{S} \times \mathcal{G} \to \mathbb{R}$.

They are trained with the following SARSA losses:

$$L^v(\theta_v) = \mathbb{E}_{(s_h, a_h, \ldots, s_{h+n}) \sim p^{\mathcal{D}}} \left[ D\left( V^\ell_{\theta_v}(s_h, s_{h+n}), Q^\ell_{\bar{\theta}_q}(s_h, a_h, s_{h+n}) \right) \right], \quad (28)$$

$$L^q(\theta_q) = \mathbb{E}_{(s_h, a_h, \ldots, s_{h+n}) \sim p^{\mathcal{D}}} \left[ D\left( Q^\ell_{\theta_q}(s_h, a_h, s_{h+n}), r(s_h, s_{h+n}) + \tilde{\gamma} V^\ell_{\theta_v}(s_{h+1}, s_{h+n}) \right) \right], \quad (29)$$

where $\tilde{\gamma}$ is the low-level discount factor defined as $\tilde{\gamma} = 1 - 1/n$. At test time, double SHARSA defines the distribution of the low-level policy $\pi^\ell_{\theta_\ell}(a \mid s, w) : \mathcal{S} \times \mathcal{G} \to \Delta(\mathcal{A})$ with rejection sampling:

$$\pi^\ell_{\theta_\ell}(s, w) \stackrel{d}{=} \underset{a_1, \ldots, a_N : a_i \sim \pi^\ell_{\beta, \theta_\ell}(a | s, w)}{\arg\max} Q^\ell_{\theta_q}(s, a_i, w). \quad (30)$$

We provide the pseudocode in Algorithm 2.

---

**Algorithm 1** SHARSA

---

**while** not converged **do**

    Sample batch $\{(s_h, a_h, \ldots, s_{h+n}, g)\}$ from $\mathcal{D}$

    ▷ Hierarchical flow BC
    Update high-level flow BC policy $\pi_\beta^h(s_{h+n} \mid s_h, g)$ with flow-matching loss (Equation (23))
    Update low-level flow BC policy $\pi_\beta^\ell(a_h \mid s_h, s_{h+n})$ with flow-matching loss (Equation (24))

    ▷ High-level ($n$-step) SARSA value learning
    Update $V^h$ to minimize $\mathbb{E}\left[D\left(V^h(s_h, g), \bar{Q}^h(s_h, s_{h+n}, g)\right)\right]$
    Update $Q^h$ to minimize $\mathbb{E}\left[D\left(Q^h(s_h, s_{h+n}, g), \sum_{i=0}^{n-1} \gamma^i r(s_{h+i}, g) + \gamma^n V^h(s_{h+n}, g)\right)\right]$

**return** $\pi(s, g)$ defined below

**function** $\pi(s, g)$

    ▷ High-level: rejection sampling
    Sample $w_1, \ldots, w_N \sim \pi_\beta^h(s, g)$
    Set $w \leftarrow \arg\max_{w_1, \ldots, w_N} Q^h(s, w_i, g)$

    ▷ Low-level: behavioral cloning
    Sample $a \sim \pi_\beta^\ell(s, w)$

    **return** $a$

---

---

**Algorithm 2** Double SHARSA

---

**while** not converged **do**

    Sample batch $\{(s_h, a_h, \ldots, s_{h+n}, g)\}$ from $\mathcal{D}$

    ▷ Hierarchical flow BC
    Update high-level flow BC policy $\pi_\beta^h(s_{h+n} \mid s_h, g)$ with flow-matching loss (Equation (23))
    Update low-level flow BC policy $\pi_\beta^\ell(a_h \mid s_h, s_{h+n})$ with flow-matching loss (Equation (24))

    ▷ High-level ($n$-step) SARSA value learning
    Update $V^h$ to minimize $\mathbb{E}\left[D\left(V^h(s_h, g), \bar{Q}^h(s_h, s_{h+n}, g)\right)\right]$
    Update $Q^h$ to minimize $\mathbb{E}\left[D\left(Q^h(s_h, s_{h+n}, g), \sum_{i=0}^{n-1} \gamma^i r(s_{h+i}, g) + \gamma^n V^h(s_{h+n}, g)\right)\right]$

    ▷ Low-level SARSA value learning
    Update $V^\ell$ to minimize $\mathbb{E}\left[D\left(V^\ell(s_h, s_{h+n}), \bar{Q}^\ell(s_h, a_h, s_{h+n})\right)\right]$
    Update $Q^\ell$ to minimize $\mathbb{E}\left[D\left(Q^\ell(s_h, a_h, s_{h+n}), r(s_h, s_{h+n}) + \gamma V^\ell(s_{h+1}, s_{h+n})\right)\right]$

**return** $\pi(s, g)$ defined below

**function** $\pi(s, g)$

    ▷ High-level: rejection sampling
    Sample $w_1, \ldots, w_N \sim \pi_\beta^h(s, g)$
    Set $w \leftarrow \arg\max_{w_1, \ldots, w_N} Q^h(s, w_i, g)$

    ▷ Low-level: rejection sampling
    Sample $a_1, \ldots, a_N \sim \pi_\beta^\ell(s, w)$
    Set $a \leftarrow \arg\max_{a_1, \ldots, a_N} Q^\ell(s, a_i, w)$

    **return** $a$

---

# F Experimental details

We implement all methods used in this work on top of the reference implementations of OGBench [75]. Each run in this work takes no more than three days on a single A5000 GPU. We provide our implementations and datasets at https://github.com/seohongpark/horizon-reduction.

## F.1 Didactic experiments

In this section, we describe additional experimental details for the experiments in Section 5.1.

**Task.** The `combination-lock` task consists of $H$ states numbered from $0$ to $H - 1$ and two discrete actions. Each state is represented by $\lceil \log_2 H \rceil$-dimensional binary vector. The ordering of the states is randomly determined by a fixed random seed, which ensures that all runs in our experiments share the same environment dynamics.

**Algorithms.** We consider $1$-step DQN and $n$-step DQN (with $n = 64$) in Section 5.1. The value losses for $1$-step DQN and $n$-step DQN are as follows:

$$
L_{1-\text{step}}(\theta) = \mathbb{E}\left[\left(Q_\theta(s_h, a_h) - r(s_h, a_h) - \max_{a_{h+1} \in \mathcal{A}} Q_{\bar{\theta}}(s_{h+1}, a_{h+1})\right)^2\right],
\tag{31}
$$

$$
L_{n-\text{step}}(\theta) = \mathbb{E}\left[\left(Q_\theta(s_h, a_h) - \sum_{i=0}^{n-1} r(s_{h+i}, a_{h+i}) - \max_{a_{h+n} \in \mathcal{A}} Q_{\bar{\theta}}(s_{h+n}, a_{h+n})\right)^2\right],
\tag{32}
$$

where the expectations are taken over consecutive state-action trajectories uniformly sampled from the dataset. We also employ double Q-learning [32] to stabilize training.

**Datasets.** We generate two types of datasets. The first is a $1$-step uniform-coverage dataset, collected by sampling state-action pairs uniformly from all possible $2H$ tuples. The second is a $64$-step uniform-coverage dataset, collected by the following procedure: first sample a state uniformly from $H$ states, and then perform either $64$ consecutive correct actions (with probability $0.5$) or $64$ consecutive incorrect actions (with probability $0.5$). Note that the former provides uniform state-action coverage for $1$-step DQN, and the latter provides uniform state-action coverage for $64$-step DQN. We use the $1$-step uniform dataset for $1$-step DQN and the $64$-step uniform dataset for $64$-step DQN. Since $1$-step DQN works worse on the $64$-step uniform dataset (Figure 15) and $64$-step DQN is incompatible with the $1$-step uniform dataset, this setup provides a fair comparison of the maximum possible performance of the two algorithms, with the dataset factor marginalized out.

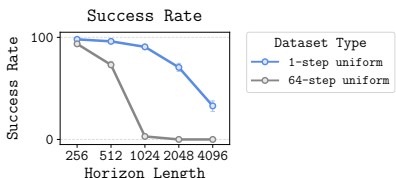

Figure 15: **$1$-step DQN on two datasets.**

**Metrics.** We train each agent for 5M gradient steps and evaluate every 100K steps. We measure three metrics: success rate, TD error, and Q error. The success rate is measured by rolling out the deterministic policy induced by the learned Q function, averaged over all evaluation epochs. The TD error is measured by the critic loss (Equations (31) and (32)), averaged over steps on and after 4M. The Q error is measured by the difference between the predicted Q values and the ground-truth Q values (*i.e.*, the negative of the remaining steps to the goal), evaluated at the final epoch.

We provide the full list of hyperparameters in Table 2.

## F.2 OGBench experiments

**Tasks.** We use three existing tasks and one new task from OGBench [75]: `humanoidmaze-giant`, `puzzle-4x5`, `puzzle-4x6`, and `cube-octuple`. In the cube domain, we extend the most challenging existing task, `cube-quadruple` (with $4$ cubes), to create a new task, `cube-octuple` (with $8$ cubes), to further challenge the agents. All these tasks are state-based and goal-conditioned. We employ the `oraclerep` variants from OGBench, which provide ground-truth goal representations (*e.g.*, in `cube`, the goal is defined only by the cube positions, not including the agent's proprioceptive states). This helps eliminate confounding factors related to goal representation learning. For the `cube-double` task used in Figure 10, we exclude the swapping task (`task4`) from the evaluation

goals (Figure 19), as we found that this task requires a non-trivial degree of distributional generalization. We refer to Figures 16 to 21 for illustrations of the evaluation goals, where the goal images for existing tasks are adopted from Park et al. [75].

**Datasets.** On each of these tasks, we generate a 1B-sized dataset using the original data-generation script provided by OGBench. The `cube` and `puzzle` datasets consist of length-1000 trajectories, and the `humanoidmaze` dataset consists of length-4000 trajectories, as in the original datasets. These datasets are collected by scripted policies that perform random tasks with a certain degree of noise. In `humanoidmaze`, the agent repeatedly reaches random positions using a (noisy) expert low-level controller; in `cube`, the agent repeatedly picks a random cube and places it in a random position; in `puzzle`, the agent repeatedly presses buttons in an arbitrary order. Notably, these datasets are collected in an unsupervised, task-agnostic manner (*i.e.*, in the "play"-style [62]). In other words, the data-collection scripts are *not* aware of the evaluation goals.

**Methods and hyperparameters.** We generally follow the original implementations, hyperparameters, and evaluation protocols of Park et al. [75]. We train each offline RL algorithm for 5M gradient steps (2.5M steps for simpler tasks in Figure 10) and evaluate every 250K steps. At each evaluation epoch, we measure the success rate of the agent using 15 rollouts on each of the 5 (4 for `cube-double`) evaluation goals. For data-scaling plots, we compute the average success rate over the last three evaluation epochs (*i.e.*, 4.5M, 4.75M, and 5M steps), following Park et al. [75].

The hyperparameters (in particular, the degree of behavioral regularization) of each algorithm are individually tuned on each task based on the largest 1B datasets. We provide the full list of hyperparameters in Tables 3 and 4, where we abbreviate $n$-step SAC+BC as $n$-SAC+BC and double SHARSA as DSHARSA.

# G  Result tables

We provide result tables in Tables 5 and 6, where standard deviations are denoted by the "±" sign. In the tables, we abbreviate flow BC as FBC, hierarchical flow BC as HFBC, $n$-step SAC+BC as $n$-SAC+BC, and double SHARSA as DSHARSA. We highlight values at or above 95% of the best performance in bold, following Park et al. [75].

Table 2: **Hyperparameters for didactic experiments.**

| Hyperparameter | Value |
|---|---|
| Gradient steps | 5M |
| Optimizer | Adam [46] |
| Learning rate | 0.0003 |
| Batch size | 512 |
| MLP size | $[512, 512, 512]$ |
| Nonlinearity | GELU [34] |
| Layer normalization | True |
| Target network update rate | 0.005 |
| Discount factor $\gamma$ | 1 |
| Horizon reduction factor $n$ | 64 |

Table 3: **Common hyperparameters for OGBench experiments.**

| Hyperparameter | Value |
|---|---|
| Gradient steps | 5M (default), 2.5M (`cube-double`, `puzzle-4x4`) |
| Optimizer | Adam [46] |
| Learning rate | 0.0003 |
| Batch size | 1024 |
| MLP size | $[1024, 1024, 1024, 1024]$ |
| Nonlinearity | GELU [34] |
| Layer normalization | True |
| Target network update rate | 0.005 |
| Discount factor $\gamma$ | 0.999 (default), 0.99 (`cube-double`, `puzzle-4x4`) |
| Flow steps | 10 |
| Horizon reduction factor $n$ | 50 (`cube`, `humanoidmaze`), 25 (`puzzle`) |
| Expectile $\kappa$ (IQL) | 0.9 |
| Expectile $\kappa$ (HIQL) | 0.5 (`cube`, `humanoidmaze`), 0.7 (`puzzle`) |
| Value representation dimensionality $k$ (CRL) | 1024 |
| Goal representation dimensionality $k$ (HIQL) | 128 |
| Double Q aggregation (SAC+BC, SHARSA, FQL) | $\min(Q_1, Q_2)$ (`cube`, `puzzle`), $(Q_1 + Q_2)/2$ (`humanoidmaze`) |
| Value loss type (SAC+BC, SHARSA, FQL) | Binary cross entropy |
| Actor $(p_{\mathrm{cur}}^{\mathcal{D}}, p_{\mathrm{geom}}^{\mathcal{D}}, p_{\mathrm{traj}}^{\mathcal{D}}, p_{\mathrm{rand}}^{\mathcal{D}})$ ratio (BC) | $(0, 1, 0, 0)$ |
| Actor $(p_{\mathrm{cur}}^{\mathcal{D}}, p_{\mathrm{geom}}^{\mathcal{D}}, p_{\mathrm{traj}}^{\mathcal{D}}, p_{\mathrm{rand}}^{\mathcal{D}})$ ratio (others) | $(0, 1, 0, 0)$ (`cube`), $(0, 0.5, 0, 0.5)$ (`puzzle`), $(0, 0, 1, 0)$ (`humanoidmaze`) |
| Value $(p_{\mathrm{cur}}^{\mathcal{D}}, p_{\mathrm{geom}}^{\mathcal{D}}, p_{\mathrm{traj}}^{\mathcal{D}}, p_{\mathrm{rand}}^{\mathcal{D}})$ ratio (CRL) | $(0, 1, 0, 0)$ |
| Value $(p_{\mathrm{cur}}^{\mathcal{D}}, p_{\mathrm{geom}}^{\mathcal{D}}, p_{\mathrm{traj}}^{\mathcal{D}}, p_{\mathrm{rand}}^{\mathcal{D}})$ ratio (others) | $(0.2, 0, 0.5, 0.3)$ |
| Policy extraction hyperparameters | Table 4 |

Table 4: **Policy extraction hyperparameters for OGBench experiments.**

| Task | IQL $\alpha$ | CRL $\alpha$ | SAC+BC $\alpha$ | FQL $\alpha$ | n-SAC+BC $\alpha$ | HIQL $\alpha$ | SHARSA $N$ | DSHARSA $N$ |
|---|---|---|---|---|---|---|---|---|
| `cube-octuple` | 10 | 3 | 10 | 3 | 0.1 | 10 | 32 | 32 |
| `puzzle-4x5` | 1 | 1 | 0.3 | 3 | 0.1 | 3 | 32 | 32 |
| `puzzle-4x6` | 1 | 1 | 0.3 | 3 | 0.1 | 3 | 32 | 32 |
| `humanoidmaze-giant` | 0.3 | 0.3 | 0.1 | 3 | 0.03 | 3 | 32 | 32 |
| `cube-double` | 3 | 10 | 1 | - | - | - | - | - |
| `puzzle-4x4` | 1 | 3 | 0.3 | - | - | - | - | - |

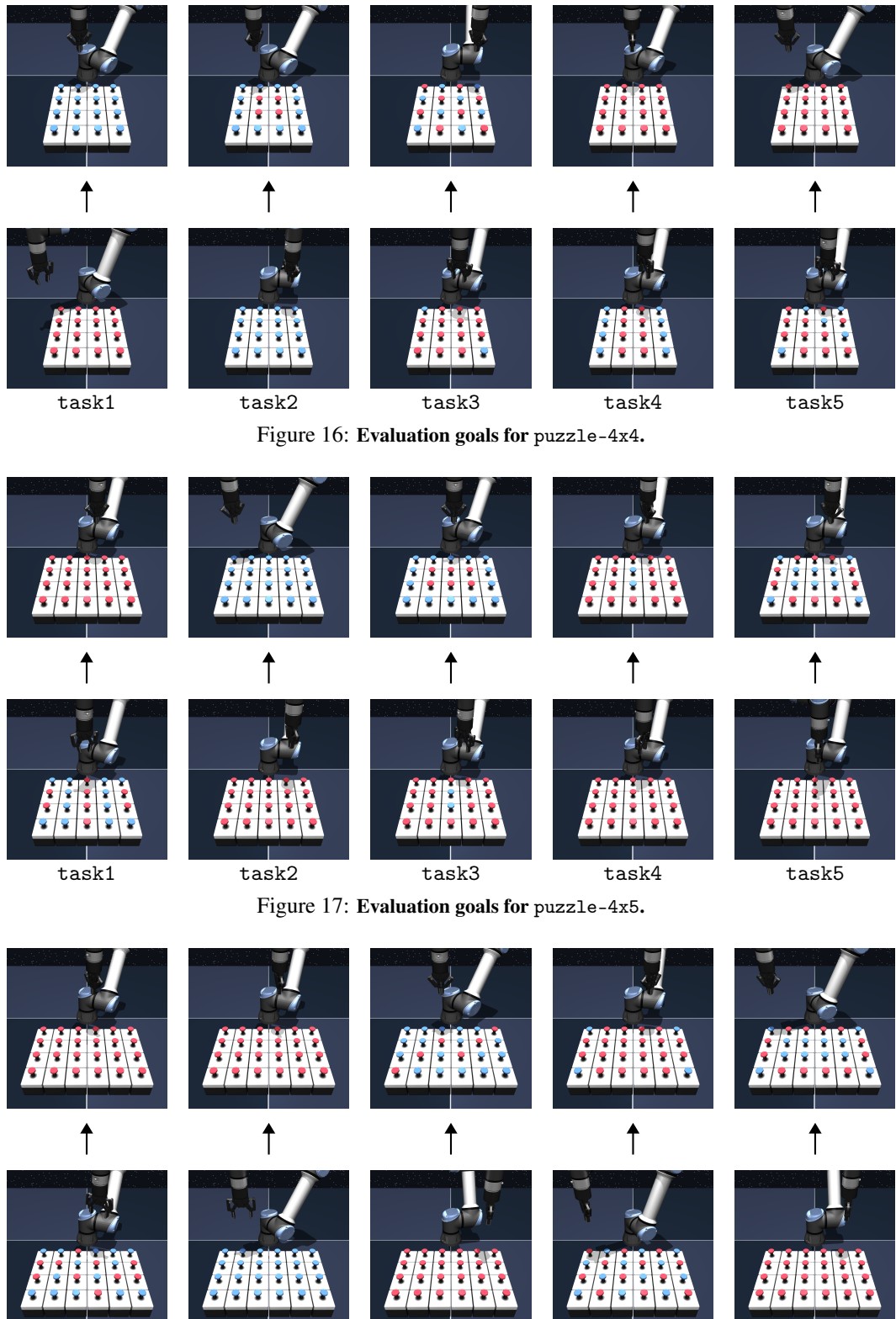

Figure 16: **Evaluation goals for** `puzzle-4x4`.

Figure 17: **Evaluation goals for** `puzzle-4x5`.

Figure 18: **Evaluation goals for** `puzzle-4x6`.

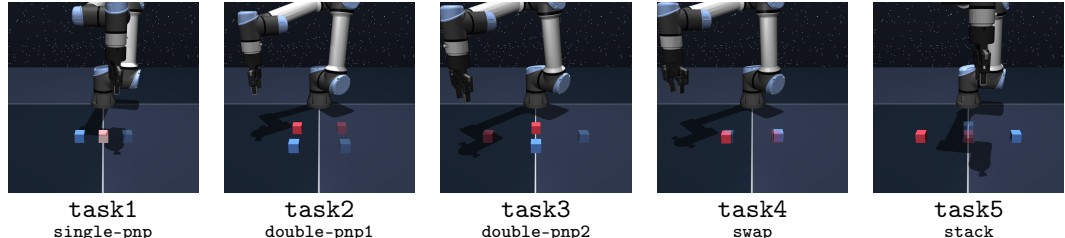

Figure 19: **Evaluation goals for** `cube-double`. As stated in Appendix F.2, `task4` is omitted from our evaluation.



Figure 20: **Evaluation goals for** `cube-octuple`.

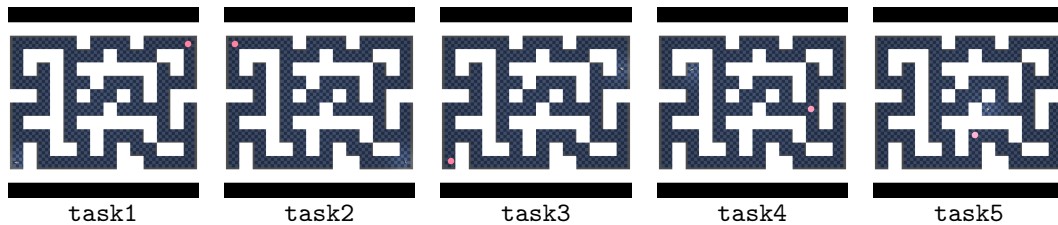

Figure 21: **Evaluation goals for** `humanoidmaze-giant`.

Table 5: **Full results (at 1M epoch).**

| Environment | Dataset Size | Task | FBC | IQL | CRL | SAC+BC | n-SAC+BC | HFBC | HIQL | SHARSA | DSHARSA |
|---|---|---|---|---|---|---|---|---|---|---|---|
| cube-octuple-play-oraclerep-v0 | 1M | task1 | 0 ±0 | 0 ±0 | 0 ±0 | 0 ±0 | 0 ±0 | 0 ±0 | 0 ±0 | 0 ±0 | 0 ±0 |
| | | task2 | 0 ±0 | 0 ±0 | 0 ±0 | 0 ±0 | 0 ±0 | 0 ±0 | 0 ±0 | 0 ±0 | 0 ±0 |
| | | task3 | 0 ±0 | 0 ±0 | 0 ±0 | 0 ±0 | 0 ±0 | 0 ±0 | 0 ±0 | 0 ±0 | 0 ±0 |
| | | task4 | 0 ±0 | 0 ±0 | 0 ±0 | 0 ±0 | 0 ±0 | 0 ±0 | 0 ±0 | 0 ±0 | 0 ±0 |
| | | task5 | 0 ±0 | 0 ±0 | 0 ±0 | 0 ±0 | 0 ±0 | 0 ±0 | 0 ±0 | 0 ±0 | 0 ±0 |
| | | overall | 0 ±0 | 0 ±0 | 0 ±0 | 0 ±0 | 0 ±0 | 0 ±0 | 0 ±0 | 0 ±0 | 0 ±0 |
| | 10M | task1 | 0 ±0 | 0 ±0 | 0 ±0 | 0 ±0 | 0 ±0 | 60 ±17 | 3 ±7 | 85 ±11 | 35 ±24 |
| | | task2 | 0 ±0 | 0 ±0 | 0 ±0 | 0 ±0 | 0 ±0 | 2 ±3 | 0 ±0 | 2 ±3 | 0 ±0 |
| | | task3 | 0 ±0 | 0 ±0 | 0 ±0 | 0 ±0 | 0 ±0 | 3 ±4 | 0 ±0 | 5 ±3 | 0 ±0 |
| | | task4 | 0 ±0 | 0 ±0 | 0 ±0 | 0 ±0 | 0 ±0 | 0 ±0 | 0 ±0 | 0 ±0 | 0 ±0 |
| | | task5 | 0 ±0 | 0 ±0 | 0 ±0 | 0 ±0 | 0 ±0 | 0 ±0 | 0 ±0 | 0 ±0 | 0 ±0 |
| | | overall | 0 ±0 | 0 ±0 | 0 ±0 | 0 ±0 | 0 ±0 | 13 ±4 | 1 ±1 | 18 ±3 | 7 ±5 |
| | 100M | task1 | 0 ±0 | 0 ±0 | 0 ±0 | 0 ±0 | 0 ±0 | 85 ±8 | 3 ±4 | 88 ±6 | 35 ±22 |
| | | task2 | 0 ±0 | 0 ±0 | 0 ±0 | 0 ±0 | 0 ±0 | 2 ±3 | 0 ±0 | 5 ±10 | 2 ±3 |
| | | task3 | 0 ±0 | 0 ±0 | 0 ±0 | 0 ±0 | 0 ±0 | 7 ±9 | 0 ±0 | 3 ±7 | 0 ±0 |
| | | task4 | 0 ±0 | 0 ±0 | 0 ±0 | 0 ±0 | 0 ±0 | 0 ±0 | 0 ±0 | 0 ±0 | 0 ±0 |
| | | task5 | 0 ±0 | 0 ±0 | 0 ±0 | 0 ±0 | 0 ±0 | 0 ±0 | 0 ±0 | 0 ±0 | 0 ±0 |
| | | overall | 0 ±0 | 0 ±0 | 0 ±0 | 0 ±0 | 0 ±0 | 19 ±4 | 1 ±1 | 19 ±3 | 7 ±4 |
| | 1B | task1 | 0 ±0 | 0 ±0 | 0 ±0 | 0 ±0 | 0 ±0 | 87 ±8 | 3 ±7 | 93 ±5 | 17 ±12 |
| | | task2 | 0 ±0 | 0 ±0 | 0 ±0 | 0 ±0 | 0 ±0 | 2 ±3 | 0 ±0 | 2 ±3 | 0 ±0 |
| | | task3 | 0 ±0 | 0 ±0 | 0 ±0 | 0 ±0 | 0 ±0 | 10 ±9 | 0 ±0 | 3 ±4 | 0 ±0 |
| | | task4 | 0 ±0 | 0 ±0 | 0 ±0 | 0 ±0 | 0 ±0 | 0 ±0 | 0 ±0 | 0 ±0 | 0 ±0 |
| | | task5 | 0 ±0 | 0 ±0 | 0 ±0 | 0 ±0 | 0 ±0 | 0 ±0 | 0 ±0 | 0 ±0 | 0 ±0 |
| | | overall | 0 ±0 | 0 ±0 | 0 ±0 | 0 ±0 | 0 ±0 | 20 ±3 | 1 ±1 | 20 ±1 | 3 ±2 |
| puzzle-4x5-play-oraclerep-v0 | 1M | task1 | 0 ±0 | 75 ±6 | 0 ±0 | 77 ±23 | 0 ±0 | 0 ±0 | 0 ±0 | 0 ±0 | 2 ±3 |
| | | task2 | 0 ±0 | 0 ±0 | 0 ±0 | 0 ±0 | 0 ±0 | 0 ±0 | 0 ±0 | 0 ±0 | 0 ±0 |
| | | task3 | 0 ±0 | 0 ±0 | 0 ±0 | 0 ±0 | 0 ±0 | 0 ±0 | 0 ±0 | 0 ±0 | 0 ±0 |
| | | task4 | 0 ±0 | 0 ±0 | 0 ±0 | 0 ±0 | 0 ±0 | 0 ±0 | 0 ±0 | 0 ±0 | 0 ±0 |
| | | task5 | 0 ±0 | 0 ±0 | 0 ±0 | 0 ±0 | 0 ±0 | 0 ±0 | 0 ±0 | 0 ±0 | 0 ±0 |
| | | overall | 0 ±0 | 15 ±1 | 0 ±0 | 15 ±5 | 0 ±0 | 0 ±0 | 0 ±0 | 0 ±0 | 0 ±1 |
| | 10M | task1 | 0 ±0 | 97 ±7 | 0 ±0 | 93 ±9 | 63 ±19 | 17 ±9 | 22 ±11 | 98 ±3 | 100 ±0 |
| | | task2 | 0 ±0 | 0 ±0 | 0 ±0 | 0 ±0 | 10 ±13 | 3 ±4 | 2 ±3 | 95 ±6 | 98 ±3 |
| | | task3 | 0 ±0 | 0 ±0 | 0 ±0 | 0 ±0 | 0 ±0 | 0 ±0 | 0 ±0 | 75 ±27 | 65 ±8 |
| | | task4 | 0 ±0 | 0 ±0 | 0 ±0 | 0 ±0 | 2 ±3 | 0 ±0 | 0 ±0 | 65 ±19 | 78 ±8 |
| | | task5 | 0 ±0 | 0 ±0 | 0 ±0 | 0 ±0 | 2 ±3 | 0 ±0 | 0 ±0 | 50 ±22 | 52 ±17 |
| | | overall | 0 ±0 | 19 ±1 | 0 ±0 | 19 ±2 | 15 ±2 | 4 ±2 | 5 ±3 | 77 ±14 | 79 ±5 |
| | 100M | task1 | 0 ±0 | 98 ±3 | 0 ±0 | 95 ±10 | 73 ±20 | 15 ±13 | 28 ±25 | 100 ±0 | 100 ±0 |
| | | task2 | 0 ±0 | 0 ±0 | 0 ±0 | 0 ±0 | 15 ±18 | 2 ±3 | 2 ±3 | 100 ±0 | 100 ±0 |
| | | task3 | 0 ±0 | 0 ±0 | 0 ±0 | 0 ±0 | 0 ±0 | 0 ±0 | 0 ±0 | 97 ±4 | 72 ±18 |
| | | task4 | 0 ±0 | 0 ±0 | 0 ±0 | 0 ±0 | 8 ±8 | 0 ±0 | 0 ±0 | 92 ±6 | 83 ±4 |
| | | task5 | 0 ±0 | 0 ±0 | 0 ±0 | 0 ±0 | 0 ±0 | 0 ±0 | 0 ±0 | 68 ±13 | 42 ±28 |
| | | overall | 0 ±0 | 20 ±1 | 0 ±0 | 19 ±2 | 19 ±4 | 3 ±2 | 6 ±5 | 91 ±4 | 79 ±8 |
| | 1B | task1 | 0 ±0 | 100 ±0 | 7 ±5 | 95 ±3 | 70 ±21 | 18 ±10 | 38 ±26 | 100 ±0 | 100 ±0 |
| | | task2 | 0 ±0 | 0 ±0 | 0 ±0 | 0 ±0 | 28 ±29 | 0 ±0 | 0 ±0 | 98 ±3 | 100 ±0 |
| | | task3 | 0 ±0 | 0 ±0 | 0 ±0 | 0 ±0 | 0 ±0 | 0 ±0 | 0 ±0 | 93 ±8 | 95 ±10 |
| | | task4 | 0 ±0 | 0 ±0 | 0 ±0 | 0 ±0 | 2 ±3 | 0 ±0 | 0 ±0 | 98 ±3 | 93 ±0 |
| | | task5 | 0 ±0 | 0 ±0 | 0 ±0 | 0 ±0 | 2 ±3 | 0 ±0 | 0 ±0 | 75 ±3 | 93 ±5 |
| | | overall | 0 ±0 | 20 ±0 | 1 ±1 | 19 ±1 | 20 ±9 | 4 ±2 | 8 ±5 | 93 ±3 | 96 ±3 |
| puzzle-4x6-play-oraclerep-v0 | 1M | task1 | 0 ±0 | 42 ±3 | 0 ±0 | 20 ±19 | 0 ±0 | 0 ±0 | 0 ±0 | 0 ±0 | 0 ±0 |
| | | task2 | 0 ±0 | 22 ±8 | 0 ±0 | 3 ±7 | 0 ±0 | 0 ±0 | 0 ±0 | 0 ±0 | 0 ±0 |
| | | task3 | 0 ±0 | 0 ±0 | 0 ±0 | 0 ±0 | 0 ±0 | 0 ±0 | 0 ±0 | 0 ±0 | 0 ±0 |
| | | task4 | 0 ±0 | 0 ±0 | 0 ±0 | 0 ±0 | 0 ±0 | 0 ±0 | 0 ±0 | 0 ±0 | 0 ±0 |
| | | task5 | 0 ±0 | 0 ±0 | 0 ±0 | 0 ±0 | 0 ±0 | 0 ±0 | 0 ±0 | 0 ±0 | 0 ±0 |
| | | overall | 0 ±0 | 13 ±2 | 0 ±0 | 5 ±3 | 0 ±0 | 0 ±0 | 0 ±0 | 0 ±0 | 0 ±0 |
| | 10M | task1 | 0 ±0 | 88 ±3 | 0 ±0 | 72 ±10 | 17 ±7 | 3 ±4 | 10 ±13 | 100 ±0 | 100 ±0 |
| | | task2 | 0 ±0 | 8 ±10 | 0 ±0 | 0 ±0 | 7 ±5 | 2 ±3 | 2 ±3 | 53 ±24 | 68 ±14 |
| | | task3 | 0 ±0 | 0 ±0 | 0 ±0 | 0 ±0 | 0 ±0 | 0 ±0 | 0 ±0 | 47 ±13 | 45 ±17 |
| | | task4 | 0 ±0 | 0 ±0 | 0 ±0 | 0 ±0 | 0 ±0 | 0 ±0 | 0 ±0 | 35 ±11 | 48 ±25 |
| | | task5 | 0 ±0 | 0 ±0 | 0 ±0 | 0 ±0 | 0 ±0 | 0 ±0 | 0 ±0 | 0 ±0 | 0 ±0 |
| | | overall | 0 ±0 | 19 ±3 | 0 ±0 | 14 ±2 | 5 ±2 | 1 ±1 | 2 ±2 | 47 ±1 | 52 ±6 |
| | 100M | task1 | 0 ±0 | 78 ±21 | 0 ±0 | 67 ±36 | 10 ±9 | 8 ±10 | 40 ±28 | 100 ±0 | 100 ±0 |
| | | task2 | 0 ±0 | 0 ±0 | 0 ±0 | 0 ±0 | 5 ±3 | 2 ±3 | 8 ±13 | 40 ±22 | 55 ±26 |
| | | task3 | 0 ±0 | 0 ±0 | 0 ±0 | 0 ±0 | 0 ±0 | 2 ±3 | 0 ±0 | 65 ±15 | 68 ±13 |
| | | task4 | 0 ±0 | 0 ±0 | 0 ±0 | 0 ±0 | 0 ±0 | 0 ±0 | 0 ±0 | 62 ±6 | 60 ±8 |
| | | task5 | 0 ±0 | 0 ±0 | 0 ±0 | 0 ±0 | 0 ±0 | 0 ±0 | 0 ±0 | 0 ±0 | 0 ±0 |
| | | overall | 0 ±0 | 16 ±4 | 0 ±0 | 13 ±7 | 3 ±2 | 2 ±2 | 10 ±8 | 53 ±4 | 57 ±8 |
| | 1B | task1 | 0 ±0 | 87 ±9 | 0 ±0 | 48 ±39 | 17 ±12 | 7 ±5 | 37 ±26 | 100 ±0 | 100 ±0 |
| | | task2 | 0 ±0 | 0 ±0 | 0 ±0 | 5 ±10 | 8 ±6 | 2 ±3 | 3 ±7 | 62 ±30 | 60 ±14 |
| | | task3 | 0 ±0 | 0 ±0 | 0 ±0 | 0 ±0 | 8 ±17 | 0 ±0 | 0 ±0 | 65 ±15 | 70 ±4 |
| | | task4 | 0 ±0 | 0 ±0 | 0 ±0 | 0 ±0 | 0 ±0 | 0 ±0 | 0 ±0 | 52 ±17 | 67 ±11 |
| | | task5 | 0 ±0 | 0 ±0 | 0 ±0 | 0 ±0 | 0 ±0 | 0 ±0 | 0 ±0 | 0 ±0 | 0 ±0 |
| | | overall | 0 ±0 | 17 ±2 | 0 ±0 | 11 ±8 | 7 ±6 | 2 ±1 | 8 ±5 | 56 ±9 | 59 ±5 |
| humanoidmaze-giant-navigate-oraclerep-v0 | 1M | task1 | 0 ±0 | 0 ±0 | 28 ±18 | 2 ±3 | 0 ±0 | 5 ±3 | 2 ±3 | 33 ±9 | 52 ±15 |
| | | task2 | 3 ±4 | 5 ±3 | 30 ±17 | 5 ±3 | 0 ±0 | 7 ±8 | 13 ±0 | 37 ±4 | 40 ±5 |
| | | task3 | 0 ±0 | 0 ±0 | 15 ±6 | 3 ±7 | 10 ±20 | 8 ±6 | 7 ±5 | 17 ±9 | 37 ±14 |
| | | task4 | 0 ±0 | 3 ±4 | 32 ±29 | 0 ±0 | 3 ±4 | 12 ±6 | 0 ±0 | 47 ±14 | 60 ±20 |
| | | task5 | 3 ±7 | 3 ±4 | 38 ±16 | 0 ±0 | 43 ±39 | 23 ±12 | 10 ±13 | 82 ±8 | 83 ±16 |
| | | overall | 1 ±1 | 2 ±1 | 29 ±15 | 2 ±1 | 11 ±9 | 11 ±4 | 6 ±2 | 43 ±3 | 54 ±7 |
| | 10M | task1 | 0 ±0 | 2 ±3 | 92 ±8 | 3 ±4 | 58 ±28 | 10 ±4 | 8 ±6 | 30 ±12 | 32 ±15 |
| | | task2 | 2 ±3 | 5 ±6 | 87 ±13 | 5 ±3 | 90 ±9 | 15 ±18 | 48 ±23 | 53 ±20 | 78 ±11 |
| | | task3 | 0 ±0 | 3 ±7 | 85 ±13 | 7 ±0 | 75 ±11 | 20 ±9 | 12 ±6 | 37 ±4 | 60 ±16 |
| | | task4 | 0 ±0 | 5 ±6 | 83 ±25 | 3 ±4 | 80 ±12 | 12 ±6 | 20 ±19 | 40 ±17 | 40 ±5 |
| | | task5 | 3 ±4 | 2 ±3 | 92 ±8 | 0 ±0 | 97 ±4 | 37 ±9 | 23 ±9 | 95 ±6 | 93 ±5 |
| | | overall | 1 ±1 | 3 ±1 | 88 ±13 | 4 ±1 | 80 ±9 | 19 ±4 | 22 ±7 | 51 ±5 | 61 ±5 |
| | 100M | task1 | 0 ±0 | 0 ±0 | 67 ±45 | 7 ±5 | 58 ±18 | 13 ±5 | 10 ±12 | 22 ±18 | 52 ±18 |
| | | task2 | 3 ±4 | 10 ±7 | 70 ±48 | 15 ±18 | 87 ±8 | 17 ±9 | 40 ±22 | 43 ±22 | 50 ±19 |
| | | task3 | 0 ±0 | 5 ±3 | 70 ±34 | 13 ±16 | 85 ±11 | 22 ±17 | 33 ±14 | 23 ±19 | 42 ±15 |
| | | task4 | 0 ±0 | 2 ±3 | 67 ±45 | 0 ±0 | 82 ±11 | 13 ±0 | 12 ±10 | 40 ±14 | 45 ±6 |
| | | task5 | 3 ±4 | 3 ±4 | 75 ±46 | 0 ±0 | 98 ±3 | 38 ±15 | 48 ±23 | 87 ±18 | 95 ±6 |
| | | overall | 1 ±1 | 4 ±2 | 70 ±43 | 7 ±4 | 82 ±5 | 21 ±5 | 29 ±8 | 43 ±6 | 57 ±6 |
| | 1B | task1 | 0 ±0 | 0 ±0 | 52 ±38 | 5 ±3 | 68 ±13 | 10 ±7 | 13 ±14 | 28 ±3 | 43 ±16 |
| | | task2 | 0 ±0 | 3 ±7 | 63 ±46 | 12 ±3 | 88 ±10 | 12 ±6 | 32 ±18 | 55 ±17 | 70 ±16 |
| | | task3 | 0 ±0 | 5 ±6 | 68 ±46 | 8 ±6 | 77 ±12 | 20 ±12 | 23 ±18 | 20 ±5 | 43 ±12 |
| | | task4 | 0 ±0 | 2 ±3 | 57 ±41 | 2 ±3 | 77 ±12 | 7 ±5 | 10 ±4 | 53 ±12 | 40 ±13 |
| | | task5 | 0 ±0 | 3 ±4 | 68 ±46 | 0 ±0 | 98 ±3 | 45 ±3 | 35 ±14 | 82 ±16 | 97 ±4 |
| | | overall | 0 ±0 | 3 ±4 | 62 ±42 | 5 ±0 | 82 ±3 | 19 ±5 | 23 ±13 | 48 ±5 | 59 ±7 |

Table 6: **Full results (averaged over 4.5M, 4.75M, and 5M epochs).**

| Environment | Dataset Size | Task | FBC | IQL | CRL | SAC+BC | n-SAC+BC | HFBC | HIQL | SHARSA | DSHARSA |
|---|---|---|---|---|---|---|---|---|---|---|---|
| cube-octuple-play-oraclerep-v0 | 1M | task1 | $\mathbf{0}_{\pm0}$ | $\mathbf{0}_{\pm0}$ | $\mathbf{0}_{\pm0}$ | $\mathbf{0}_{\pm0}$ | $\mathbf{0}_{\pm0}$ | $\mathbf{0}_{\pm0}$ | $\mathbf{0}_{\pm0}$ | $\mathbf{0}_{\pm0}$ | $\mathbf{0}_{\pm0}$ |
| | | task2 | $\mathbf{0}_{\pm0}$ | $\mathbf{0}_{\pm0}$ | $\mathbf{0}_{\pm0}$ | $\mathbf{0}_{\pm0}$ | $\mathbf{0}_{\pm0}$ | $\mathbf{0}_{\pm0}$ | $\mathbf{0}_{\pm0}$ | $\mathbf{0}_{\pm0}$ | $\mathbf{0}_{\pm0}$ |
| | | task3 | $\mathbf{0}_{\pm0}$ | $\mathbf{0}_{\pm0}$ | $\mathbf{0}_{\pm0}$ | $\mathbf{0}_{\pm0}$ | $\mathbf{0}_{\pm0}$ | $\mathbf{0}_{\pm0}$ | $\mathbf{0}_{\pm0}$ | $\mathbf{0}_{\pm0}$ | $\mathbf{0}_{\pm0}$ |
| | | task4 | $\mathbf{0}_{\pm0}$ | $\mathbf{0}_{\pm0}$ | $\mathbf{0}_{\pm0}$ | $\mathbf{0}_{\pm0}$ | $\mathbf{0}_{\pm0}$ | $\mathbf{0}_{\pm0}$ | $\mathbf{0}_{\pm0}$ | $\mathbf{0}_{\pm0}$ | $\mathbf{0}_{\pm0}$ |
| | | task5 | $\mathbf{0}_{\pm0}$ | $\mathbf{0}_{\pm0}$ | $\mathbf{0}_{\pm0}$ | $\mathbf{0}_{\pm0}$ | $\mathbf{0}_{\pm0}$ | $\mathbf{0}_{\pm0}$ | $\mathbf{0}_{\pm0}$ | $\mathbf{0}_{\pm0}$ | $\mathbf{0}_{\pm0}$ |
| | | overall | $\mathbf{0}_{\pm0}$ | $\mathbf{0}_{\pm0}$ | $\mathbf{0}_{\pm0}$ | $\mathbf{0}_{\pm0}$ | $\mathbf{0}_{\pm0}$ | $\mathbf{0}_{\pm0}$ | $\mathbf{0}_{\pm0}$ | $\mathbf{0}_{\pm0}$ | $\mathbf{0}_{\pm0}$ |
| | 10M | task1 | $0_{\pm0}$ | $0_{\pm0}$ | $0_{\pm0}$ | $0_{\pm0}$ | $0_{\pm0}$ | $46_{\pm11}$ | $1_{\pm1}$ | $\mathbf{79}_{\pm3}$ | $42_{\pm6}$ |
| | | task2 | $0_{\pm0}$ | $0_{\pm0}$ | $0_{\pm0}$ | $0_{\pm0}$ | $0_{\pm0}$ | $\mathbf{1}_{\pm1}$ | $0_{\pm0}$ | $\mathbf{1}_{\pm1}$ | $\mathbf{1}_{\pm2}$ |
| | | task3 | $0_{\pm0}$ | $0_{\pm0}$ | $0_{\pm0}$ | $0_{\pm0}$ | $0_{\pm0}$ | $\mathbf{2}_{\pm2}$ | $0_{\pm0}$ | $\mathbf{2}_{\pm2}$ | $0_{\pm0}$ |
| | | task4 | $\mathbf{0}_{\pm0}$ | $\mathbf{0}_{\pm0}$ | $\mathbf{0}_{\pm0}$ | $\mathbf{0}_{\pm0}$ | $\mathbf{0}_{\pm0}$ | $\mathbf{0}_{\pm0}$ | $\mathbf{0}_{\pm0}$ | $\mathbf{0}_{\pm0}$ | $\mathbf{0}_{\pm0}$ |
| | | task5 | $\mathbf{0}_{\pm0}$ | $\mathbf{0}_{\pm0}$ | $\mathbf{0}_{\pm0}$ | $\mathbf{0}_{\pm0}$ | $\mathbf{0}_{\pm0}$ | $\mathbf{0}_{\pm0}$ | $\mathbf{0}_{\pm0}$ | $\mathbf{0}_{\pm0}$ | $\mathbf{0}_{\pm0}$ |
| | | overall | $0_{\pm0}$ | $0_{\pm0}$ | $0_{\pm0}$ | $0_{\pm0}$ | $0_{\pm0}$ | $10_{\pm3}$ | $0_{\pm0}$ | $\mathbf{16}_{\pm1}$ | $9_{\pm1}$ |
| | 100M | task1 | $0_{\pm0}$ | $0_{\pm0}$ | $0_{\pm0}$ | $0_{\pm0}$ | $0_{\pm0}$ | $87_{\pm4}$ | $33_{\pm14}$ | $\mathbf{97}_{\pm1}$ | $81_{\pm6}$ |
| | | task2 | $0_{\pm0}$ | $0_{\pm0}$ | $0_{\pm0}$ | $0_{\pm0}$ | $0_{\pm0}$ | $11_{\pm6}$ | $3_{\pm2}$ | $\mathbf{24}_{\pm7}$ | $15_{\pm4}$ |
| | | task3 | $0_{\pm0}$ | $0_{\pm0}$ | $0_{\pm0}$ | $0_{\pm0}$ | $0_{\pm0}$ | $23_{\pm9}$ | $3_{\pm3}$ | $\mathbf{58}_{\pm6}$ | $14_{\pm10}$ |
| | | task4 | $0_{\pm0}$ | $0_{\pm0}$ | $0_{\pm0}$ | $0_{\pm0}$ | $0_{\pm0}$ | $\mathbf{1}_{\pm2}$ | $0_{\pm0}$ | $0_{\pm0}$ | $0_{\pm0}$ |
| | | task5 | $0_{\pm0}$ | $0_{\pm0}$ | $0_{\pm0}$ | $0_{\pm0}$ | $0_{\pm0}$ | $\mathbf{6}_{\pm3}$ | $0_{\pm0}$ | $1_{\pm1}$ | $0_{\pm0}$ |
| | | overall | $0_{\pm0}$ | $0_{\pm0}$ | $0_{\pm0}$ | $0_{\pm0}$ | $0_{\pm0}$ | $26_{\pm3}$ | $8_{\pm4}$ | $\mathbf{36}_{\pm1}$ | $22_{\pm2}$ |
| | 1B | task1 | $0_{\pm0}$ | $0_{\pm0}$ | $0_{\pm0}$ | $0_{\pm0}$ | $0_{\pm0}$ | $84_{\pm9}$ | $56_{\pm20}$ | $\mathbf{99}_{\pm1}$ | $75_{\pm9}$ |
| | | task2 | $0_{\pm0}$ | $0_{\pm0}$ | $0_{\pm0}$ | $0_{\pm0}$ | $0_{\pm0}$ | $18_{\pm5}$ | $\mathbf{28}_{\pm29}$ | $1_{\pm2}$ | $1_{\pm2}$ |
| | | task3 | $0_{\pm0}$ | $0_{\pm0}$ | $0_{\pm0}$ | $0_{\pm0}$ | $0_{\pm0}$ | $\mathbf{29}_{\pm4}$ | $23_{\pm25}$ | $6_{\pm2}$ | $3_{\pm1}$ |
| | | task4 | $0_{\pm0}$ | $0_{\pm0}$ | $0_{\pm0}$ | $0_{\pm0}$ | $0_{\pm0}$ | $0_{\pm0}$ | $\mathbf{1}_{\pm1}$ | $0_{\pm0}$ | $0_{\pm0}$ |
| | | task5 | $0_{\pm0}$ | $0_{\pm0}$ | $0_{\pm0}$ | $0_{\pm0}$ | $0_{\pm0}$ | $\mathbf{4}_{\pm4}$ | $2_{\pm2}$ | $0_{\pm0}$ | $0_{\pm0}$ |
| | | overall | $0_{\pm0}$ | $0_{\pm0}$ | $0_{\pm0}$ | $0_{\pm0}$ | $0_{\pm0}$ | $\mathbf{27}_{\pm4}$ | $22_{\pm15}$ | $21_{\pm1}$ | $16_{\pm2}$ |
| puzzle-4x5-play-oraclerep-v0 | 1M | task1 | $0_{\pm0}$ | $38_{\pm9}$ | $0_{\pm0}$ | $\mathbf{87}_{\pm9}$ | $0_{\pm0}$ | $0_{\pm0}$ | $0_{\pm0}$ | $1_{\pm1}$ | $1_{\pm1}$ |
| | | task2 | $\mathbf{0}_{\pm0}$ | $\mathbf{0}_{\pm0}$ | $\mathbf{0}_{\pm0}$ | $\mathbf{0}_{\pm0}$ | $\mathbf{0}_{\pm0}$ | $\mathbf{0}_{\pm0}$ | $\mathbf{0}_{\pm0}$ | $\mathbf{0}_{\pm0}$ | $\mathbf{0}_{\pm0}$ |
| | | task3 | $\mathbf{0}_{\pm0}$ | $\mathbf{0}_{\pm0}$ | $\mathbf{0}_{\pm0}$ | $\mathbf{0}_{\pm0}$ | $\mathbf{0}_{\pm0}$ | $\mathbf{0}_{\pm0}$ | $\mathbf{0}_{\pm0}$ | $\mathbf{0}_{\pm0}$ | $\mathbf{0}_{\pm0}$ |
| | | task4 | $0_{\pm0}$ | $\mathbf{1}_{\pm1}$ | $0_{\pm0}$ | $0_{\pm0}$ | $0_{\pm0}$ | $0_{\pm0}$ | $0_{\pm0}$ | $0_{\pm0}$ | $0_{\pm0}$ |
| | | task5 | $\mathbf{0}_{\pm0}$ | $\mathbf{0}_{\pm0}$ | $\mathbf{0}_{\pm0}$ | $\mathbf{0}_{\pm0}$ | $\mathbf{0}_{\pm0}$ | $\mathbf{0}_{\pm0}$ | $\mathbf{0}_{\pm0}$ | $\mathbf{0}_{\pm0}$ | $\mathbf{0}_{\pm0}$ |
| | | overall | $0_{\pm0}$ | $8_{\pm2}$ | $0_{\pm0}$ | $\mathbf{17}_{\pm2}$ | $0_{\pm0}$ | $0_{\pm0}$ | $0_{\pm0}$ | $0_{\pm0}$ | $0_{\pm0}$ |
| | 10M | task1 | $4_{\pm0}$ | $\mathbf{99}_{\pm1}$ | $0_{\pm0}$ | $97_{\pm4}$ | $83_{\pm7}$ | $13_{\pm3}$ | $4_{\pm4}$ | $\mathbf{100}_{\pm0}$ | $\mathbf{100}_{\pm0}$ |
| | | task2 | $0_{\pm0}$ | $0_{\pm0}$ | $0_{\pm0}$ | $0_{\pm0}$ | $56_{\pm17}$ | $0_{\pm0}$ | $0_{\pm0}$ | $99_{\pm1}$ | $\mathbf{100}_{\pm0}$ |
| | | task3 | $0_{\pm0}$ | $0_{\pm0}$ | $0_{\pm0}$ | $0_{\pm0}$ | $21_{\pm10}$ | $0_{\pm0}$ | $0_{\pm0}$ | $96_{\pm4}$ | $96_{\pm2}$ |
| | | task4 | $0_{\pm0}$ | $0_{\pm0}$ | $0_{\pm0}$ | $0_{\pm0}$ | $26_{\pm17}$ | $0_{\pm0}$ | $0_{\pm0}$ | $95_{\pm4}$ | $\mathbf{98}_{\pm1}$ |
| | | task5 | $0_{\pm0}$ | $0_{\pm0}$ | $0_{\pm0}$ | $0_{\pm0}$ | $17_{\pm9}$ | $0_{\pm0}$ | $0_{\pm0}$ | $83_{\pm7}$ | $\mathbf{88}_{\pm5}$ |
| | | overall | $1_{\pm0}$ | $20_{\pm0}$ | $0_{\pm0}$ | $19_{\pm1}$ | $40_{\pm9}$ | $3_{\pm1}$ | $1_{\pm1}$ | $95_{\pm2}$ | $\mathbf{97}_{\pm2}$ |
| | 100M | task1 | $1_{\pm1}$ | $\mathbf{100}_{\pm0}$ | $73_{\pm5}$ | $94_{\pm5}$ | $96_{\pm4}$ | $26_{\pm6}$ | $44_{\pm30}$ | $\mathbf{100}_{\pm0}$ | $\mathbf{100}_{\pm0}$ |
| | | task2 | $1_{\pm1}$ | $0_{\pm0}$ | $0_{\pm0}$ | $0_{\pm0}$ | $81_{\pm16}$ | $1_{\pm1}$ | $1_{\pm1}$ | $99_{\pm1}$ | $\mathbf{100}_{\pm0}$ |
| | | task3 | $0_{\pm0}$ | $0_{\pm0}$ | $0_{\pm0}$ | $0_{\pm0}$ | $66_{\pm22}$ | $1_{\pm1}$ | $0_{\pm0}$ | $\mathbf{99}_{\pm2}$ | $94_{\pm6}$ |
| | | task4 | $0_{\pm0}$ | $0_{\pm0}$ | $0_{\pm0}$ | $0_{\pm0}$ | $54_{\pm12}$ | $0_{\pm0}$ | $0_{\pm0}$ | $93_{\pm7}$ | $\mathbf{96}_{\pm2}$ |
| | | task5 | $0_{\pm0}$ | $0_{\pm0}$ | $0_{\pm0}$ | $0_{\pm0}$ | $61_{\pm20}$ | $0_{\pm0}$ | $0_{\pm0}$ | $\mathbf{87}_{\pm4}$ | $79_{\pm12}$ |
| | | overall | $0_{\pm0}$ | $20_{\pm0}$ | $15_{\pm1}$ | $19_{\pm1}$ | $71_{\pm12}$ | $6_{\pm2}$ | $9_{\pm6}$ | $\mathbf{96}_{\pm2}$ | $94_{\pm3}$ |
| | 1B | task1 | $1_{\pm1}$ | $97_{\pm4}$ | $96_{\pm2}$ | $79_{\pm15}$ | $96_{\pm5}$ | $26_{\pm8}$ | $48_{\pm26}$ | $\mathbf{100}_{\pm0}$ | $\mathbf{100}_{\pm0}$ |
| | | task2 | $1_{\pm1}$ | $0_{\pm0}$ | $0_{\pm0}$ | $0_{\pm0}$ | $74_{\pm8}$ | $1_{\pm2}$ | $1_{\pm2}$ | $\mathbf{100}_{\pm0}$ | $\mathbf{100}_{\pm0}$ |
| | | task3 | $0_{\pm0}$ | $0_{\pm0}$ | $0_{\pm0}$ | $0_{\pm0}$ | $71_{\pm12}$ | $0_{\pm0}$ | $0_{\pm0}$ | $\mathbf{99}_{\pm2}$ | $98_{\pm2}$ |
| | | task4 | $0_{\pm0}$ | $0_{\pm0}$ | $0_{\pm0}$ | $0_{\pm0}$ | $52_{\pm22}$ | $0_{\pm0}$ | $0_{\pm0}$ | $93_{\pm2}$ | $\mathbf{100}_{\pm0}$ |
| | | task5 | $0_{\pm0}$ | $0_{\pm0}$ | $0_{\pm0}$ | $0_{\pm0}$ | $64_{\pm10}$ | $1_{\pm1}$ | $0_{\pm0}$ | $89_{\pm6}$ | $\mathbf{94}_{\pm6}$ |
| | | overall | $0_{\pm0}$ | $19_{\pm1}$ | $19_{\pm0}$ | $16_{\pm3}$ | $71_{\pm9}$ | $5_{\pm2}$ | $10_{\pm6}$ | $96_{\pm2}$ | $\mathbf{98}_{\pm1}$ |
| puzzle-4x6-play-oraclerep-v0 | 1M | task1 | $0_{\pm0}$ | $2_{\pm2}$ | $0_{\pm0}$ | $\mathbf{44}_{\pm21}$ | $0_{\pm0}$ | $0_{\pm0}$ | $0_{\pm0}$ | $0_{\pm0}$ | $0_{\pm0}$ |
| | | task2 | $0_{\pm0}$ | $0_{\pm0}$ | $0_{\pm0}$ | $\mathbf{1}_{\pm1}$ | $0_{\pm0}$ | $0_{\pm0}$ | $0_{\pm0}$ | $0_{\pm0}$ | $0_{\pm0}$ |
| | | task3 | $\mathbf{0}_{\pm0}$ | $\mathbf{0}_{\pm0}$ | $\mathbf{0}_{\pm0}$ | $\mathbf{0}_{\pm0}$ | $\mathbf{0}_{\pm0}$ | $\mathbf{0}_{\pm0}$ | $\mathbf{0}_{\pm0}$ | $\mathbf{0}_{\pm0}$ | $\mathbf{0}_{\pm0}$ |
| | | task4 | $\mathbf{0}_{\pm0}$ | $\mathbf{0}_{\pm0}$ | $\mathbf{0}_{\pm0}$ | $\mathbf{0}_{\pm0}$ | $\mathbf{0}_{\pm0}$ | $\mathbf{0}_{\pm0}$ | $\mathbf{0}_{\pm0}$ | $\mathbf{0}_{\pm0}$ | $\mathbf{0}_{\pm0}$ |
| | | task5 | $\mathbf{0}_{\pm0}$ | $\mathbf{0}_{\pm0}$ | $\mathbf{0}_{\pm0}$ | $\mathbf{0}_{\pm0}$ | $\mathbf{0}_{\pm0}$ | $\mathbf{0}_{\pm0}$ | $\mathbf{0}_{\pm0}$ | $\mathbf{0}_{\pm0}$ | $\mathbf{0}_{\pm0}$ |
| | | overall | $0_{\pm0}$ | $0_{\pm0}$ | $0_{\pm0}$ | $\mathbf{9}_{\pm4}$ | $0_{\pm0}$ | $0_{\pm0}$ | $0_{\pm0}$ | $0_{\pm0}$ | $0_{\pm0}$ |
| | 10M | task1 | $1_{\pm1}$ | $91_{\pm7}$ | $0_{\pm0}$ | $64_{\pm39}$ | $53_{\pm19}$ | $7_{\pm5}$ | $2_{\pm2}$ | $\mathbf{100}_{\pm0}$ | $\mathbf{100}_{\pm0}$ |
| | | task2 | $0_{\pm0}$ | $2_{\pm3}$ | $0_{\pm0}$ | $6_{\pm8}$ | $13_{\pm11}$ | $2_{\pm2}$ | $0_{\pm0}$ | $54_{\pm11}$ | $\mathbf{62}_{\pm18}$ |
| | | task3 | $0_{\pm0}$ | $0_{\pm0}$ | $0_{\pm0}$ | $0_{\pm0}$ | $11_{\pm11}$ | $1_{\pm1}$ | $0_{\pm0}$ | $\mathbf{78}_{\pm8}$ | $73_{\pm11}$ |
| | | task4 | $0_{\pm0}$ | $0_{\pm0}$ | $0_{\pm0}$ | $0_{\pm0}$ | $1_{\pm1}$ | $0_{\pm0}$ | $0_{\pm0}$ | $50_{\pm8}$ | $\mathbf{69}_{\pm8}$ |
| | | task5 | $\mathbf{0}_{\pm0}$ | $\mathbf{0}_{\pm0}$ | $\mathbf{0}_{\pm0}$ | $\mathbf{0}_{\pm0}$ | $\mathbf{0}_{\pm0}$ | $\mathbf{0}_{\pm0}$ | $\mathbf{0}_{\pm0}$ | $\mathbf{0}_{\pm0}$ | $\mathbf{0}_{\pm0}$ |
| | | overall | $0_{\pm0}$ | $18_{\pm2}$ | $0_{\pm0}$ | $14_{\pm6}$ | $16_{\pm7}$ | $2_{\pm1}$ | $0_{\pm0}$ | $57_{\pm3}$ | $\mathbf{61}_{\pm4}$ |
| | 100M | task1 | $2_{\pm2}$ | $64_{\pm12}$ | $7_{\pm8}$ | $71_{\pm9}$ | $48_{\pm8}$ | $13_{\pm5}$ | $42_{\pm25}$ | $\mathbf{100}_{\pm0}$ | $\mathbf{100}_{\pm0}$ |
| | | task2 | $0_{\pm0}$ | $0_{\pm0}$ | $1_{\pm1}$ | $1_{\pm1}$ | $38_{\pm10}$ | $4_{\pm2}$ | $2_{\pm3}$ | $57_{\pm8}$ | $\mathbf{77}_{\pm11}$ |
| | | task3 | $0_{\pm0}$ | $0_{\pm0}$ | $0_{\pm0}$ | $0_{\pm0}$ | $19_{\pm11}$ | $1_{\pm1}$ | $1_{\pm1}$ | $89_{\pm3}$ | $\mathbf{92}_{\pm3}$ |
| | | task4 | $0_{\pm0}$ | $0_{\pm0}$ | $0_{\pm0}$ | $0_{\pm0}$ | $1_{\pm1}$ | $1_{\pm1}$ | $0_{\pm0}$ | $72_{\pm10}$ | $\mathbf{91}_{\pm2}$ |
| | | task5 | $\mathbf{0}_{\pm0}$ | $\mathbf{0}_{\pm0}$ | $\mathbf{0}_{\pm0}$ | $\mathbf{0}_{\pm0}$ | $\mathbf{0}_{\pm0}$ | $\mathbf{0}_{\pm0}$ | $\mathbf{0}_{\pm0}$ | $\mathbf{0}_{\pm0}$ | $\mathbf{0}_{\pm0}$ |
| | | overall | $0_{\pm0}$ | $13_{\pm2}$ | $2_{\pm2}$ | $14_{\pm2}$ | $21_{\pm3}$ | $4_{\pm2}$ | $9_{\pm5}$ | $64_{\pm1}$ | $\mathbf{72}_{\pm3}$ |
| | 1B | task1 | $1_{\pm1}$ | $61_{\pm26}$ | $57_{\pm13}$ | $78_{\pm7}$ | $66_{\pm13}$ | $14_{\pm5}$ | $44_{\pm17}$ | $\mathbf{100}_{\pm0}$ | $\mathbf{100}_{\pm0}$ |
| | | task2 | $1_{\pm1}$ | $0_{\pm0}$ | $3_{\pm4}$ | $0_{\pm0}$ | $76_{\pm9}$ | $4_{\pm4}$ | $5_{\pm5}$ | $\mathbf{83}_{\pm16}$ | $74_{\pm19}$ |
| | | task3 | $0_{\pm0}$ | $0_{\pm0}$ | $0_{\pm0}$ | $0_{\pm0}$ | $29_{\pm26}$ | $2_{\pm2}$ | $0_{\pm0}$ | $91_{\pm4}$ | $\mathbf{94}_{\pm4}$ |
| | | task4 | $0_{\pm0}$ | $0_{\pm0}$ | $0_{\pm0}$ | $0_{\pm0}$ | $9_{\pm15}$ | $0_{\pm0}$ | $0_{\pm0}$ | $83_{\pm5}$ | $\mathbf{93}_{\pm3}$ |
| | | task5 | $\mathbf{0}_{\pm0}$ | $\mathbf{0}_{\pm0}$ | $\mathbf{0}_{\pm0}$ | $\mathbf{0}_{\pm0}$ | $\mathbf{0}_{\pm0}$ | $\mathbf{0}_{\pm0}$ | $\mathbf{0}_{\pm0}$ | $\mathbf{0}_{\pm0}$ | $\mathbf{0}_{\pm0}$ |
| | | overall | $0_{\pm0}$ | $12_{\pm5}$ | $12_{\pm3}$ | $16_{\pm1}$ | $36_{\pm9}$ | $4_{\pm1}$ | $10_{\pm4}$ | $71_{\pm5}$ | $\mathbf{72}_{\pm4}$ |
| humanoidmaze-giant-navigate-oraclerep-v0 | 1M | task1 | $0_{\pm0}$ | $0_{\pm0}$ | $30_{\pm15}$ | $2_{\pm2}$ | $0_{\pm0}$ | $2_{\pm1}$ | $9_{\pm5}$ | $12_{\pm3}$ | $\mathbf{38}_{\pm3}$ |
| | | task2 | $1_{\pm1}$ | $2_{\pm2}$ | $39_{\pm15}$ | $12_{\pm10}$ | $2_{\pm3}$ | $7_{\pm4}$ | $28_{\pm5}$ | $16_{\pm5}$ | $\mathbf{55}_{\pm10}$ |
| | | task3 | $1_{\pm1}$ | $1_{\pm0}$ | $29_{\pm7}$ | $1_{\pm1}$ | $0_{\pm0}$ | $3_{\pm3}$ | $31_{\pm6}$ | $11_{\pm4}$ | $\mathbf{39}_{\pm8}$ |
| | | task4 | $0_{\pm0}$ | $0_{\pm0}$ | $48_{\pm26}$ | $1_{\pm1}$ | $0_{\pm0}$ | $8_{\pm1}$ | $23_{\pm13}$ | $26_{\pm10}$ | $\mathbf{50}_{\pm14}$ |
| | | task5 | $3_{\pm2}$ | $2_{\pm0}$ | $54_{\pm25}$ | $1_{\pm1}$ | $2_{\pm3}$ | $23_{\pm9}$ | $44_{\pm8}$ | $56_{\pm9}$ | $\mathbf{84}_{\pm2}$ |
| | | overall | $1_{\pm1}$ | $1_{\pm0}$ | $40_{\pm14}$ | $3_{\pm2}$ | $1_{\pm1}$ | $9_{\pm2}$ | $27_{\pm6}$ | $24_{\pm3}$ | $\mathbf{53}_{\pm2}$ |
| | 10M | task1 | $2_{\pm2}$ | $7_{\pm5}$ | $\mathbf{82}_{\pm14}$ | $16_{\pm5}$ | $71_{\pm2}$ | $11_{\pm7}$ | $51_{\pm8}$ | $55_{\pm9}$ | $74_{\pm16}$ |
| | | task2 | $13_{\pm9}$ | $23_{\pm9}$ | $81_{\pm15}$ | $49_{\pm11}$ | $83_{\pm5}$ | $19_{\pm8}$ | $75_{\pm4}$ | $58_{\pm2}$ | $\mathbf{87}_{\pm8}$ |
| | | task3 | $3_{\pm3}$ | $28_{\pm18}$ | $78_{\pm9}$ | $36_{\pm11}$ | $79_{\pm6}$ | $17_{\pm7}$ | $78_{\pm7}$ | $41_{\pm8}$ | $\mathbf{82}_{\pm7}$ |
| | | task4 | $1_{\pm2}$ | $12_{\pm13}$ | $84_{\pm11}$ | $19_{\pm6}$ | $82_{\pm4}$ | $13_{\pm6}$ | $53_{\pm6}$ | $75_{\pm9}$ | $\mathbf{85}_{\pm6}$ |
| | | task5 | $7_{\pm2}$ | $15_{\pm14}$ | $92_{\pm11}$ | $4_{\pm4}$ | $97_{\pm1}$ | $49_{\pm5}$ | $79_{\pm5}$ | $97_{\pm3}$ | $\mathbf{98}_{\pm3}$ |
| | | overall | $5_{\pm1}$ | $17_{\pm9}$ | $84_{\pm11}$ | $25_{\pm5}$ | $82_{\pm2}$ | $22_{\pm4}$ | $67_{\pm3}$ | $65_{\pm2}$ | $\mathbf{85}_{\pm2}$ |
| | 100M | task1 | $4_{\pm4}$ | $4_{\pm3}$ | $71_{\pm44}$ | $39_{\pm13}$ | $76_{\pm9}$ | $18_{\pm10}$ | $60_{\pm11}$ | $36_{\pm3}$ | $56_{\pm10}$ |
| | | task2 | $11_{\pm5}$ | $19_{\pm5}$ | $78_{\pm41}$ | $50_{\pm14}$ | $93_{\pm1}$ | $27_{\pm8}$ | $80_{\pm9}$ | $54_{\pm7}$ | $69_{\pm12}$ |
| | | task3 | $5_{\pm3}$ | $25_{\pm13}$ | $77_{\pm38}$ | $34_{\pm15}$ | $\mathbf{94}_{\pm4}$ | $23_{\pm8}$ | $79_{\pm4}$ | $31_{\pm9}$ | $60_{\pm8}$ |
| | | task4 | $6_{\pm5}$ | $14_{\pm5}$ | $79_{\pm33}$ | $42_{\pm23}$ | $\mathbf{90}_{\pm6}$ | $17_{\pm5}$ | $68_{\pm6}$ | $50_{\pm13}$ | $56_{\pm9}$ |
| | | task5 | $7_{\pm3}$ | $22_{\pm8}$ | $77_{\pm40}$ | $14_{\pm6}$ | $\mathbf{100}_{\pm0}$ | $52_{\pm8}$ | $91_{\pm5}$ | $95_{\pm5}$ | $99_{\pm1}$ |
| | | overall | $6_{\pm2}$ | $17_{\pm3}$ | $76_{\pm39}$ | $36_{\pm13}$ | $\mathbf{91}_{\pm2}$ | $27_{\pm5}$ | $76_{\pm4}$ | $53_{\pm4}$ | $68_{\pm4}$ |
| | 1B | task1 | $3_{\pm3}$ | $8_{\pm3}$ | $63_{\pm40}$ | $39_{\pm10}$ | $\mathbf{81}_{\pm9}$ | $14_{\pm7}$ | $59_{\pm8}$ | $40_{\pm9}$ | $63_{\pm11}$ |
| | | task2 | $10_{\pm3}$ | $18_{\pm5}$ | $67_{\pm40}$ | $44_{\pm16}$ | $\mathbf{94}_{\pm3}$ | $21_{\pm7}$ | $81_{\pm2}$ | $42_{\pm6}$ | $78_{\pm6}$ |
| | | task3 | $5_{\pm6}$ | $21_{\pm5}$ | $69_{\pm34}$ | $36_{\pm9}$ | $\mathbf{87}_{\pm5}$ | $22_{\pm7}$ | $73_{\pm1}$ | $34_{\pm3}$ | $65_{\pm12}$ |
| | | task4 | $1_{\pm1}$ | $9_{\pm7}$ | $71_{\pm41}$ | $32_{\pm5}$ | $\mathbf{93}_{\pm7}$ | $21_{\pm4}$ | $56_{\pm6}$ | $47_{\pm5}$ | $64_{\pm14}$ |
| | | task5 | $12_{\pm6}$ | $16_{\pm5}$ | $77_{\pm35}$ | $14_{\pm13}$ | $98_{\pm3}$ | $50_{\pm7}$ | $89_{\pm2}$ | $94_{\pm5}$ | $\mathbf{99}_{\pm1}$ |
| | | overall | $6_{\pm2}$ | $14_{\pm2}$ | $69_{\pm38}$ | $33_{\pm9}$ | $\mathbf{91}_{\pm2}$ | $26_{\pm2}$ | $72_{\pm2}$ | $51_{\pm4}$ | $74_{\pm6}$ |

