# OpenReview forum: "Horizon Reduction Makes RL Scalable"
_NeurIPS.cc/2025/Conference — NeurIPS 2025 spotlight_

### Official Review · Reviewer_dG9E · 2025-07-01

**Clarity:** 3
**Significance:** 3
**Originality:** 3
**Rating:** 5
**Confidence:** 3

**Summary:**

This manuscript investigates the scalability of offline RL algorithms and demonstrates, through experiments on datasets with a scale of up to one billion samples, that many standard offline RL methods struggle to perform well on complex problems. The authors identify the "horizon" as a key limiting factor for scaling offline RL and provide empirical evidence to support this claim. To address this issue, they propose a simple yet effective method—SHARSA—that shortens the horizon and achieves both strong asymptotic performance and favorable scalability characteristics.

**Questions:**

My main concern lies in the apparent disconnect between the choice of goal-conditioned RL tasks and the analysis of the horizon curse in this manuscript. Specifically:

-	The manuscript emphasizes that error accumulation in value learning is a key cause of the horizon curse. However, if I understand correctly, this phenomenon is not necessarily unique to goal-conditioned RL. Offline RL more generally suffers from extrapolation errors due to distributional shift (OOD issues), and these errors also accumulate during the Bellman backup process — a point that has been discussed in prior work such as [1].

-	In Lines 269–270, the authors state that in goal-conditioned settings, the mapping between optimal actions and distant goals can be highly complex, due to its dependence on the topology of the entire state space. I wonder whether this issue also applies to non-goal-conditioned settings, and if so, what the underlying reasons are.

In summary, I am unclear whether the horizon curse discussed in this manuscript is a general challenge in offline RL, or one that is specific to goal-conditioned RL tasks. If it is the latter, I suggest the authors revise the framing of the work accordingly, including the title. If it is the former, then I would strongly encourage the authors to provide additional discussion and experiments on standard offline RL benchmarks such as Mujoco, beyond goal-conditioned tasks alone.

[1] Reining generalization in offline reinforcement learning via representation distinction. NeurIPS, 2023.

**Ethical Concerns:**

["NO or VERY MINOR ethics concerns only"]

**Final Justification:**

Most of my concerns have been well addressed, and I have decided to raise my score to a 5.

**Limitations:**

yes

**Paper Formatting Concerns:**

-	Some figures exceed the page margins, for example, Figure 5.

-	There is excessive use of italics in the manuscript; for instance, the word "and" on line 298 is unnecessarily italicized.

**Quality:**

3

**Strengths And Weaknesses:**

# Strengths

-	The manuscript is well-motivated and addresses a novel and important topic.

-	The writing is clear, well-structured, and easy to follow.

-	The experimental validation is thorough and detailed, adding to the credibility of the findings.

# Weaknesses

-	The experimental setup is motivated by the statement: "In particular, we are interested in the capabilities of offline RL algorithms to solve challenging tasks that require complex, long-horizon sequential decision-making given enough data." However, it remains unclear whether the curse of horizon is a universally limiting factor in offline RL. See Question 1 for more details.

---

> ### Author Rebuttal · Authors · 2025-07-29
>
> Thank you for the highly detailed review and constructive feedback about this work. We especially appreciate your question about non-goal-conditioned settings. We have conducted additional experiments on $4$ standard (reward-based) offline RL tasks to show that our claim applies to general offline RL as well. Please find our detailed response below.
>
> * **"I am unclear whether the horizon curse discussed in this manuscript is a general challenge in offline RL, or one that is specific to goal-conditioned RL tasks"**
>
> Thanks for raising this point. In the paper, we mainly focus on offline goal-conditioned RL, as this problem setting naturally provides diverse challenges in offline RL (multi-task, long-horizon reasoning, stitching, etc.). However, we believe our claim ("horizon reduction makes offline RL scalable") is not limited to offline goal-conditioned RL, and generally applies to (reward-based) offline RL settings too. For example, our analysis in Section 5.1 is done in standard reward-based offline RL.
>
> To further support this point, we additionally conducted experiments with $4$ standard **reward-based** offline RL tasks with `singletask` environments (`task1`, to be specific) provided by OGBench. These tasks are reward-based as in D4RL, but are more challenging than standard D4RL tasks (we would like to note that D4RL Gym MuJoCo tasks are generally already saturated [1, 2]). On these tasks, we evaluated SARSA (i.e., IQL with expectile $\kappa = 0.5$), IQL (with $\kappa = 0.7$), SAC+BC, n-step SAC+BC (value-only horizon reduction), and SHARSA (value and policy horizon reduction). Additionally, we consider an IQL variant of SHARSA (SHARSA with $\kappa = 0.7$, a variant introduced in Appendix C), which can be particularly helpful as these `singletask` tasks have (much) higher suboptimality due to the absence of hindsight relabeling.
>
> |                               | $\texttt{SARSA}$   | $\texttt{IQL}$     | $\texttt{SAC+BC}$ | $\texttt{n-SAC+BC}$  | $\texttt{SHARSA}$ $(\kappa=0.5)$   | $\texttt{SHARSA}$ $(\kappa=0.7)$   |
> |:--------------------------------------------------|:-------------------|:-------------------|:--------------------|:----------------------------|:-----------------------------------|:-----------------------------------|
> | $\texttt{Horizon Reduction Type}$ | - | - | - | $\texttt{Value}$ | $\texttt{Value \\& Policy}$ | $\texttt{Value \\& Policy}$ |
> | | | | | | |
> | $\texttt{cube-quadruple-play-singletask}$         | $0 {\tiny \pm 0}$  | $7 {\tiny \pm 6}$  | $0 {\tiny \pm 0}$   | $0 {\tiny \pm 0}$           | $50 {\tiny \pm 9}$                 | $\mathbf{60} {\tiny \pm 8}$        |
> | $\texttt{puzzle-4x5-play-singletask}$             | $4 {\tiny \pm 3}$  | $24 {\tiny \pm 7}$ | $0 {\tiny \pm 0}$   | $0 {\tiny \pm 0}$           | $\mathbf{94} {\tiny \pm 3}$        | $\mathbf{94} {\tiny \pm 2}$        |
> | $\texttt{puzzle-4x6-play-singletask}$             | $2 {\tiny \pm 2}$  | $5 {\tiny \pm 2}$  | $0 {\tiny \pm 0}$   | $0 {\tiny \pm 0}$           | $\mathbf{14} {\tiny \pm 6}$        | $\mathbf{14} {\tiny \pm 7}$        |
> | $\texttt{humanoidmaze-giant-navigate-singletask}$ | $0 {\tiny \pm 0}$  | $2 {\tiny \pm 2}$  | $44 {\tiny \pm 35}$ | $\mathbf{88} {\tiny \pm 3}$ | $26 {\tiny \pm 4}$                 | $\mathbf{87} {\tiny \pm 3}$        |
>
> The table above shows the results on four tasks (at 1M steps, with 4 seeds and standard deviations), with 100M- (`cube`) or 1B- (others) sized datasets. The results suggest that both value and policy horizon reduction techniques can significantly improve the performance of reward-based standard offline RL as well. Specifically, they show that value horizon reduction is particularly helpful (and is enough) in `humanoidmaze`, but both value and policy horizon reductions are important in the other three tasks (`cube` and `puzzle`).
>
> Following the reviewer's suggestion, we will revise the manuscript and discuss relevant works to show that our claim applies to general long-horizon offline RL tasks (not limited to goal-conditioned settings) as well.
>
> * **Would policy horizon reduction help in non-goal-conditioned settings as well?**
>
> The results above (especially `cube` and `puzzle`) show that having hierarchical policies often substantially helps improve performance in non-goal-conditioned tasks as well. The reason is similar: the mapping between states and optimal actions can be complex in long-horizon tasks, and hierarchical policies can help reduce this complexity by decomposing this potentially complex mapping into smaller pieces.
>
> * **"The manuscript emphasizes that error accumulation in value learning is a key cause of the horizon curse. However, if I understand correctly, this phenomenon is not necessarily unique to goal-conditioned RL. Offline RL more generally suffers from extrapolation errors due to distributional shift (OOD issues), and these errors also accumulate during the Bellman backup process — a point that has been discussed in prior work such as [1]."**
>
> Indeed, and as we discussed above, error accumulation in TD learning is not unique to goal-conditioned RL (and we didn't intend to imply this!). We will further clarify this point in the final version of the paper to prevent potential misunderstanding.
>
> * **Formatting concerns**
>
> We will fix/revise them. Thanks!
>
> ---
>
> We would like to thank you again for raising important questions about the applicability of our claim to non-goal-conditioned settings. We believe the additional results and clarifications have strengthened the paper. Please let us know if you have any additional concerns or questions.
>
> [1] Tarasov et al., Revisiting the Minimalist Approach to Offline Reinforcement Learning, NeurIPS 2023. \
> [2] Rafailov et al., D5RL: Diverse Datasets for Data-Driven Deep Reinforcement Learning, RLC 2024.

---

### Official Review · Reviewer_qjCd · 2025-07-02

**Clarity:** 3
**Significance:** 3
**Originality:** 2
**Rating:** 5
**Confidence:** 3

**Summary:**

This paper investigates an important issue in offline reinforcement learning (RL): the failure of current algorithms to scale effectively to long-horizon, complex tasks, even when provided with massive datasets (up to 1 billion transitions). The authors' central contribution is to identify and empirically validate the "curse of horizon" as a primary problem. They decompose this curse into two key issues: (1) the accumulation of value-estimation errors in temporal difference (TD) learning, and (2) the explosion in policy complexity for tasks with distant goals. As a solution, the paper systematically demonstrates that methods incorporating "horizon reduction" can effectively mitigate these issues and unlock scalability.

**Questions:**

See weakness.

**Ethical Concerns:**

["NO or VERY MINOR ethics concerns only"]

**Final Justification:**

The authors have addressed my concern about the limited effectiveness of reducing the policy horizon and have promised to clarify the potentially ambiguous term “horizon reduction.” I am therefore raising my score to 5.

**Limitations:**

Yes. They are upfront about the dependence on data quality (Section 7), the idealized nature of the state-based environments, and the fact that even their proposed method does not achieve perfect performance.

**Quality:**

3

**Strengths And Weaknesses:**

### Strengths

The empirical work in this paper is a major strength. The use of datasets up to 1 billion samples is rare in previous research and provides evidence for their claims about performance saturation. The design of the "combination-lock" toy problem is an excellent example of a controlled experiment that provides clear evidence for the "bias accumulation" hypothesis in value learning. The choice of difficult, long-horizon tasks from OGBench demonstrates the practical relevance of the problem.

### Weaknesses

The primary weakness of this paper lies in its limited technical novelty. The proposed "Horizon Reduction" technique for value learning is essentially the well-established n-step target method. Similarly, the approach for policy learning introduces a standard hierarchical framework. Both are familiar techniques that have been widely used in the field.

The term "Horizon Reduction" is potentially misleading. In the context of Markov Decision Processes (MDPs), the "horizon" typically refers to the length of a trajectory (H). The use of n-step returns for the value function target does not actually shorten this trajectory horizon but rather adjusts the Bellman backup mechanism. A more precise term would be "n-step value bootstrapping" or similar.

Moreover, the empirical results suggest that the performance gains stem predominantly from the application of n-step targets. As seen in Figure 9, a simpler SAC+BC baseline already performs quite well, which questions the severity of the "curse of horizon" in the policy learning. This raises doubts about the necessity of introducing a hierarchical policy, a modification that adds significant complexity and may hinder the method's scalability.

### Overall Assessment
In summary, while the technical contributions are incremental, the paper's large-scale experiments are valuable and offer insights for future research in offline reinforcement learning. For this reason, my overall assessment is positive.

---

> ### Author Rebuttal · Authors · 2025-07-29
>
> Thank you for the highly detailed review and constructive feedback about this work. Please find our response below.
>
> * **Limited technical novelty**
>
> As the reviewer points out, the main goal of this work is not to propose an entirely novel technique (L306-308). Instead, we aim to provide *new insights* based on our systematic scaling analysis. Through our scaling study, we identified the horizon as a fundamental bottleneck in offline RL scaling, and proposed a simple, scalable recipe that leads to better asymptotic performance.
>
> To the best of our knowledge, this paper is the first to perform a 1 billion-scale data-scaling analysis in offline RL (with orders of magnitude larger datasets than the closest prior analysis work [1]), showing that horizon reduction leads to better *scalability*. We would like to note that most previous works that study horizon reduction (n-step returns, hierarchical RL, etc.) have mainly focused on sample efficiency, performance, or exploration, rather than scalability (L96-102 in Section 2).
>
> We believe our new, actionable perspective in offline RL scaling (with our proposed recipe, SHARSA) aligns well with the broader goals stated in the [NeurIPS 2025 Reviewer Guidelines](https://neurips.cc/Conferences/2025/ReviewerGuidelines):
>
> > [O]riginality does not necessarily require introducing an entirely new method. Rather, a work that provides novel insights by evaluating existing methods, or demonstrates improved efficiency, fairness, etc. is also equally valuable.
>
> * **The performance gains seem to mainly come from n-step targets**
>
> We would like to note that this is true only on one task, `humanoidmaze-giant`. In other tasks, for example, in `cube-octuple` (Figure 9), we were not able to achieve non-trivial performance only with value horizon reduction (i.e., n-step targets), whereas policy horizon reduction (i.e., hierarchical policies) led to substantial performance improvement. In `puzzle-4x5` and `puzzle-4x6` (Figure 9), we show that policy horizon reduction further improves the performance of n-step targets, as evidenced by the gap between n-step SAC+BC (which only reduces the value horizon) and SHARSA (which reduces *both* the value and policy horizons). We will further emphasize this point in the final draft.
>
> * **"The term horizon reduction is potentially misleading"**
>
> Thank you for the feedback! We used "horizon reduction" as a general umbrella term to encompass *both* n-step TD targets (i.e., value horizon reduction) and hierarchical policies (i.e., policy horizon reduction). While n-step bootstrapping does not reduce the true horizon, it reduces the effective number of Bellman recursion steps within an episode, which we believe can be broadly interpreted as (effective) horizon reduction.
>
> To prevent potential confusion, we will further clarify the term "(effective) horizon reduction" in the final draft. We hope this further clarification addresses the reviewer's concern, but please do let us know if you have better alternatives/suggestions for terms that can capture both n-step bootstrapping and hierarchical policies.
>
> ---
>
> We would like to thank you again for raising important questions about this work, and we believe the additional clarifications have strengthened the paper. Please let us know if you have any additional concerns or questions.
>
> [1] Park et al., Is Value Learning Really the Main Bottleneck in Offline RL?, NeurIPS 2024.

---

> > ### Comment · Reviewer_qjCd · 2025-08-01
> >
> > Thanks for your detailed rebuttal. I agree that the large-scale experiments in the paper provide valuable new insights, which is also why I gave a positive score. I also acknowledge that it is indeed difficult to find a single term that can capture both n-step bootstrapping and hierarchical policies. I believe the authors can keep using the term horizon reduction, but it would be helpful to clarify its meaning in the paper to avoid potential misunderstanding.
> >
> > Lastly, I'm curious whether there is a priority in terms of importance between n-step targets and hierarchical policies. At this point, I personally find n-step targets to be more crucial, as they seem to offer stronger benefits with less need for fine-tuning engineering hyperparameters compared to hierarchical approaches.

---

> > > ### Author Response · Authors · 2025-08-01
> > >
> > > Thanks for the quick response! Following the suggestion, we will further clarify the term horizon reduction in the final version of the paper.
> > >
> > > Regarding n-step targets vs. hierarchical policies -- While n-step returns (value horizon reduction) are indeed already quite effective, our results in Figure 9 show that *additionally* reducing the policy horizon can often lead to substantial further performance improvement. In `puzzle-4x6`, SHARSA (value and policy horizon reduction) achieves almost twice the success rate of n-step SAC+BC (value horizon reduction), and we see similar large gaps in `cube-octuple` and `puzzle-4x5` too. Moreover, sometimes policy horizon reduction *alone* can be highly effective, as shown in the difference between hierarchical FBC and FBC in `cube-octuple` (Figure 9). This tells us that we want to reduce *both* the value and policy horizons, which is the main motivation behind SHARSA.
> > >
> > > Finally, we would like to note that hierarchical policies do not always require additional hyperparameters (other than the horizon length, which already exists in n-step returns). In fact, we designed SHARSA with this precise goal in mind -- SHARSA is a minimalistic method that doesn't require extensive hyperparameter tuning (L60). As shown in Table 3 in Appendix, SHARSA uses a single, fixed hyperparameter value of $N = 32$ for all tasks, unlike other approaches, such as n-step SAC+BC.
> > >
> > > We would like to thank you again for the detailed review and quick response. Feel free to let us know whenever you have additional questions or concerns. If our response has resolved the main concerns, we'd be grateful if you would consider updating the rating.

---

> > > > ### Comment · Reviewer_qjCd · 2025-08-03
> > > >
> > > > Thank you for your detailed response—it addressed my main concerns. I’m raising my rating to 5.

---

### Official Review · Reviewer_mzHq · 2025-07-02

**Clarity:** 4
**Significance:** 4
**Originality:** 4
**Rating:** 6
**Confidence:** 4

**Summary:**

This paper presents experiments on how value horizon and policy horizon affect the ability of offline RL algorithms to meaningfully improve from additional data. They show that, depending on the task, large value horizons and/or large policy horizons essentially prevent learning, even when given access to very large data sets. Finally, a simple method that combines existing techniques for reducing these horizons is shown to dramatically improve the ability of offline RL to take advantage of additional data. The paper also promises to release a large dataset for further investigating such large-horizon effects.

**Questions:**

- How/why were the tasks chosen? Is it because they are easy to access, easy to generate data for, have an appropriate “challenge”? Some other reason?
- How sensitive do you expect your results to be w.r.t. task choice, assuming tasks are still complex enough to have large horizons?
- n-step SAC+BC seems best on maze. Do you suspect this is due to some quirk of how the hierarchies are implemented?
- SHARSA seems to beat double SHARSA under some specific conditions, and not all methods increase performance with more data monotonically. Is this within expectations given the challenges of implementing SHARSA?

**Ethical Concerns:**

["NO or VERY MINOR ethics concerns only"]

**Final Justification:**

I stand by my original statement. I think this is a very solid paper.

**Limitations:**

Yes

**Quality:**

4

**Strengths And Weaknesses:**

Strengths:

- The paper is well-written and easy to read.
- Claims are justified and/or explained and the logical argument from hypothesis to experiment is clear.
- The paper presents generalizable knowledge and is thus likely to have significant impact.
- The experiments are simple, convincing, and explained and justified clearly.
- The paper offers a potentially very useful dataset.

Weaknesses:

- There are no major weaknesses in my opinion, other than a few additional questions I have about task choice, which I understand may be hard to address given the space.

---

> ### Author Rebuttal · Authors · 2025-07-29
>
> Thank you for the highly detailed review and positive feedback about this work! Please find our response below.
>
> * **How/why were the tasks chosen? Is it because they are easy to access, easy to generate data for, have an appropriate “challenge”? Some other reason?**
>
> Our goal was to perform a controlled scaling analysis of offline RL, and thus we sought a benchmark that provides highly challenging tasks with controllable data-generation scripts. OGBench was a great example for this, as it provides both diverse, challenging (unsolved) tasks and scripted expert policies. Among OGBench tasks, we chose the most difficult task in each category (`cube-octuple`, `puzzle-{4x5, 4x6}`, and `humanoidmaze-giant`) to cover various types of challenges in offline RL.
>
> * **How sensitive do you expect your results to be w.r.t. task choice, assuming tasks are still complex enough to have large horizons?**
>
> Thanks for the question. We believe the horizon is a fundamental challenge in (offline) RL, as shown in our analysis (Section 5.1), and expect that the results generally apply to other long-horizon tasks too. In our experiments, we empirically support this claim on tasks across different domains (navigation, game-like puzzles, and manipulation).
>
> * **n-step SAC+BC seems best on maze. Do you suspect this is due to some quirk of how the hierarchies are implemented?**
>
> As the reviewer points out, n-step SAC+BC generally led to the best performance on `humanoidmaze-giant`, being slightly better than SHARSA. We believe this (relatively minor) performance difference mainly stems from the difference in policy extraction. SAC+BC uses reparameterized gradients (as in DDPG), whereas SHARSA uses rejection sampling. A prior work [1] shows that reparameterized gradients generally lead to better performance than rejection sampling, as the former utilizes first-order gradient information. While we chose to use rejection sampling in SHARSA for simplicity (as our goal with SHARSA was to provide a simple, scalable recipe, rather than achieving the best possible performance), we believe it is possible to further improve the performance of SHARSA with more advanced low-level policy extraction methods like reparameterized gradients.
>
> * **SHARSA seems to beat double SHARSA under some specific conditions, and not all methods increase performance with more data monotonically. Is this within expectations given the challenges of implementing SHARSA?**
>
> While SHARSA generally leads to the best performance among the methods we evaluated, as the reviewer points out, it does not always monotonically improve performance with more data in our experiments. We believe this is mainly due to random variation in optimality in generated datasets. For example, a 100M-sized dataset might have a larger *ratio* of relatively more optimal transitions for a specific evaluation task than a 1B-sized one, which may potentially explain the reversed trend in `cube-octuple`. However, we believe an ideal algorithm should not lead to worse performance with more data (despite the change in state-action distributions), and we will acknowledge this point as a limitation in Section 7 in the final version of the paper.
>
> ---
>
> We would like to thank you again for asking important questions about our work. We believe the additional clarifications have strengthened the paper. Please let us know if you have any additional concerns or questions.
>
> [1] Park et al., Is Value Learning Really the Main Bottleneck in Offline RL?, NeurIPS 2024.

---

> > ### Comment · Reviewer_mzHq · 2025-08-05
> > **Official Comment by Reviewer mzHq**
> >
> > Thank you for the clarifications / question answers. I have no remaining questions or concerns.

---

### Official Review · Reviewer_2ypq · 2025-07-02

**Clarity:** 4
**Significance:** 4
**Originality:** 3
**Rating:** 5
**Confidence:** 3

**Summary:**

This paper studies the scalability of standard MLP-based offline RL algorithms. Ideally, provided with a huge expert dataset that widely covers the state and action spaces, a good offline RL algorithm should be able to achieve perfect performance regardless of task difficulty. However, by increasing the size of the dataset from 1M to 1B, the author shows that current offline RL algorithms fail to achieve non-trivial performance on four challenging long-horizon OGBench tasks. Then, by demonstrating with a toy environment, Combination-Lock, the author hypothesizes that (1) the bias accumulation in TD value learning over lone horizon, and (2) the inability of direct mapping between optimal actions and distant goals in policy learning, orthogonally hinders the scalability of offline RL algorithms.  Therefore, the author discusses several horizon reduction methods that can mitigate the bias accumulation in value learning, e.g., n-step returns, or decompose the long-horizon policy extraction into a hierarchy. Moreover, the author proposes (Double) SHARSA that combines horizon reduction in both value and policy learning in a minimalist way. Experiments show that horizon reduction can effectively improve the offline RL performance as the data size scales up.

**Questions:**

1. In Line 233-235, the author describes two types of datasets used for the Combination-Lock environment, evaluates 1-step DQN and 64-step DQN on both datasets, and reports the best performance for each method. How do you perform the 64-step DQN on the dataset of uniform coverage of length-1 trajectory segments?
2. This work mainly focuses on the *MLP-based* offline RL methods. Could the author comment on the scalability of *Transformer-based* offline RL or imitation learning methods, e.g., Behavior Transformer (BeT), Vector-Quantized Behavior Transformer (VQ-BeT), or Q-Transformer? Does transformer architecture bring any advantage of scalability to those methods naturally? Will the curse of horizon hamper their scalability to the same extent as it hinders the MLP-based methods?

**Ethical Concerns:**

["NO or VERY MINOR ethics concerns only"]

**Final Justification:**

I did not identify the major weakness of this paper, and I list its positive aspects below:

- This paper raises and answers a research question that should be of interest to the RL community, as RL scalability remains underexplored compared to fields like NLP and CV.

- The paper touches on several types of goal-conditioned offline RL algorithms and extensively evaluates them across four meaningful control tasks with data scaling.

- The insight on horizon reduction is actionable, and the proposed methods and experimental protocol have the potential to inspire future work in improving offline RL scalability.

- During the rebuttal, the authors have addressed my questions accordingly.

Therefore, I recommend acceptance.

**Paper Formatting Concerns:**

No major formatting issues.

**Quality:**

4

**Strengths And Weaknesses:**

### Strengths:
- The paper is coherent, well-organized, and easy to follow. The figures are clean, informative, and contribute effectively to the overall presentation.
- The research question is clearly defined and appropriately scoped, with no overclaims. The hypotheses are supported by enough empirical results. A thorough ablation studies are provided in the main body and in the appendix, which addresses many potential concerns regarding the paper’s arguments. And the experimental details are well-documented in the appendix for reproduction.
- The research question should be of interest to the RL community, as RL scalability remains underexplored compared to fields like NLP and CV. The paper touches on several types of offline RL algorithms and evaluates them across four meaningful control tasks. The insight on horizon reduction is actionable, and the proposed methods and experimental protocol have the potential to inspire future work in improving offline RL scalability.
---
### Weaknesses:
I do not identify major weaknesses, and a few comments are listed below:
- Regarding the novelty, while the curse of horizon has been discussed in the RL literature, and the proposed method (SHARSA) builds upon existing techniques, the paper's central contribution—the connection between the curse of horizon and offline RL scalability—is both insightful and actionable. The paper has the potential to serve as a benchmark and reference point for future research in this area.
- In Line 276-277, the author borrows an analogy to Chain-of-Thought reasoning. Although I understand what it conveys, it appears a bit irrelevant and implies little to the hierarchical policy learning discussed in this section.
- In Line 321-331, the author describes the SHARSA value learning process but does not mention how horizon reduction is done here (although details are included in the appendix). This part would be more self-contained if the author briefly mentions that n-step return is used for high-level SHARSA value learning, while 1-step return is used for low-level (double) SHARSA learning.

---

> ### Author Rebuttal · Authors · 2025-07-29
>
> Thank you for the detailed review and constructive feedback about this work. Please find our response below.
>
> * **Could the author comment on the scalability of Transformer-based offline RL methods?**
>
> Thanks for this question! We also had a similar question, and have already investigated whether using Transformers instead of MLPs can improve the scalability of offline RL in Appendix B (L652-663). The results in Figure 11 show that, although replacing MLPs with Transformers improves performance to some degree on two tasks, it does *not* necessarily substantially improve scalability in general, and does not match the performance of horizon reduction techniques. This further supports our hypothesis that the horizon is a fundamental bottleneck in offline RL scaling.
>
> That being said, we believe recent advancements in policy architectures (e.g., BeT, VQ-BeT, etc.) might independently contribute to performance improvements, orthogonal to horizon reduction. Finally, while Q-Transformer shows promising initial results, our analysis suggests that the way Q-Transformer deals with the horizon (i.e., treating individual actions as separate steps) might lead to suboptimal scalability, as it effectively *expands* the horizon rather than reduces it.
>
>
> * **How do you perform the 64-step DQN on the dataset of uniform coverage of length-1 trajectory segments?**
>
> As the reviewer correctly pointed out, 64-step DQN is not applicable to the 1-step uniform dataset, so we evaluated 64-step DQN only on the 64-step uniform dataset (Appendix F.1, L857-858). On the other hand, we evaluated 1-step DQN on both the 1-step and 64-step uniform datasets and selected the best-performing one. Even though this selection procedure favors 1-step DQN, we show that 64-step DQN outperforms 1-step DQN. We will further clarify this point in the final version of the paper.
>
> * **Further details about SHARSA and double SHARSA (L321-331)**
>
> Thanks for the feedback. The difference between SHARSA and double SHARSA lies in the low-level policy: SHARSA uses behavioral cloning for the low-level policy, whereas double SHARSA uses an additional round of rejection sampling based on another (1-step, short-horizon) SARSA value function (Appendix E.3). As the camera-ready version allows one additional page, we will include detailed explanations of SHARSA and double SHARSA in the main paper of the final version.
>
> * **Analogy to chain-of-thought reasoning is less clear (L276-277)**
>
> Thanks for the feedback. We wanted to mention that hierarchical policies and chain-of-thought reasoning have a common ground in the sense that decomposing a problem into subtasks can improve performance. We will further clarify this analogy in the final version of the paper!
>
> ---
>
> We would like to thank you again for raising important questions about Transformer results and other clarification concerns, and we believe the additional clarifications have strengthened the paper. Please let us know if you have any additional concerns or questions.

---

> > ### Comment · Reviewer_2ypq · 2025-08-02
> >
> > Thank the authors for the detailed clarification. I agree with your comments on the Transformer-based offline RL algorithms. And I believe that the authors will revise the paper accordingly in terms of other points I made. I do not have further questions, and I would keep my score, as it is already suggesting acceptance.

---

### Note · Authors · 2025-08-13

We would like to thank the reviewers for their helpful feedback and discussion. As a final remark, we provide a brief summary of the major changes made in our rebuttal:

- Following Reviewer dG9E's suggestion, we performed additional experiments on 4 standard, reward-based offline RL tasks with large-scale datasets. The results show that our claim (i.e., horizon reduction improves scalability of offline RL) applies to general offline RL settings too, not limited to offline goal-conditioned RL.
- As mentioned in the responses, we will add various clarifications to the paper (e.g., further details about SHARSA, the term "horizon reduction", the importance of policy horizon reduction, etc.) to address the clarification concerns raised by the reviewers.

We hope that these experiments and clarifications have mostly addressed the issues raised during the review period.

---

### Decision · Program_Chairs · 2025-09-17

**Decision:**

Accept (spotlight)

**Comment:**

This is a well-prepared paper that studies why offline RL fails to scale on long-horizon tasks, identifies the horizon as the primary bottleneck, and shows that reducing the effective value and policy horizons improves scaling. The proposed method is minimalistic and practical, and the empirical evidence is very thorough, using datasets up to one billion transitions on challenging OGBench tasks; the paper is clearly written and easy to follow.

Reviews are overall positive. During discussion, reviewers raised questions about scope beyond goal-conditioned settings and about the novelty of the techniques. The authors ran additional experiments on standard, reward-based offline RL and clarified terminology and design choices, after which all reviewers converged to very positive recommendations. The AC weighs the careful analysis, scale of the evidence, and the clean takeaways for practice more heavily than concerns about incremental technical novelty.

This is a nice paper that presents interesting and solid results. The paper offers clear insight about scaling offline RL and backs it with convincing experiments and a simple method likely to influence future work.